# Self-organization of swimmers drives long-range fluid transport in bacterial colonies

Haoran Xu[1], Justas Dauparas[2], Debasish Das[2], Eric Lauga [2] & Yilin Wu [1]

Motile subpopulations in microbial communities are believed to be important for dispersal, quest for food, and material transport. Here, we show that motile cells in sessile colonies of peritrichously flagellated bacteria can self-organize into two adjacent, centimeter-scale motile rings surrounding the entire colony. The motile rings arise from spontaneous segregation of a homogeneous swimmer suspension that mimics a phase separation; the process is mediated by intercellular interactions and shear-induced depletion. As a result of this self-organization, cells drive fluid flows that circulate around the colony at a constant peak speed of ~30 $\mu$m s$^{-1}$, providing a stable and high-speed avenue for directed material transport at the macroscopic scale. Our findings present a unique form of bacterial self-organization that influences population structure and material distribution in colonies.

[1] Department of Physics and Shenzhen Research Institute, The Chinese University of Hong Kong, Shatin, New Territories, Hong Kong, People's Republic of China. [2] Department of Applied Mathematics and Theoretical Physics, University of Cambridge, Cambridge CB3 0WA, UK. Correspondence and requests for materials should be addressed to Y.W. (email: ylwu@cuhk.edu.hk)

Microbes commonly live in structured communities that play important roles in human health and ecology[1–5], such as biofilm aggregates during chronic infections[6], commensal microbiome in animal guts, microbial mats in river beds or ocean floors, and bacterial colonies in soils or food products[5]. The multicellular lifestyle is key to the prospering of microbes in diverse habitats as it confers high resistance to various environmental stresses[7–11]. Understanding the physiology of these structured microbial communities is not only essential to the treatment of chronic infections, but also important to industrial applications, such as bioremediation, anti-biofouling and food hygiene. In addition, as a type of active matter, microbial communities have provided model systems for the research of biological self-organization and complex fluids[12–16].

Heterogeneous populations, including motile and non-motile populations, often coexist in bacteria communities[17]. Motile subpopulations in microbial communities are believed to be important to dispersal[18,19], quest for food[20], and material transport[21]. However, except in circumstances where motile cells drive colony expansion (e.g., bacterial swarming[8,10,22]), the physiological functions of motile subpopulations in bacterial communities are largely unclear. For example, in many occasions bacterial colonies are sessile and their expansion is driven by growth rather than by cell motility, but these sessile colonies normally preserve a subpopulation of motile cells for reasons that are not well understood[17].

Colony mode of bacterial growth on solid substrates (e.g., food products and solidified nutrient agar) is a common experimental approach to study structured microbial communities[11] and is closely related to biofilm development[23,24]. Cells in a bacterial colony move in confined space surrounded by boundaries between gas, liquid, and solid phases. Interfaces and spatial confinement are known to affect motile behavior of bacteria in more artificial settings such as microfluidic systems. Bacteria swimming near liquid–solid interfaces have the tendency moving in parallel to the wall due to hydrodynamic trapping[25–30], but it was less clear how swimming bacteria behave at the gas–liquid–solid three phase interface, such as the edge of bacterial colonies. Also it was reported that confined bacterial suspension droplets of size less than ~100 micron[31,32] can self-organize into vortices, in which two concentric, counter-rotating regions of bacteria are present near the edge of the droplet. This type of bacterial self-organization has not yet been observed in the context of naturally developed colonies. Understanding how bacterial motility adapts to the physical environment in colonies may therefore have implications for active matter self-organization and for the engineering of self-assembled microfluidic systems that perform fluid pumping and cargo transport[33–38].

Here we sought to examine the behavior of motile cell populations in bacterial colonies and to explore their potential physiological functions. In routinely cultured bacterial colonies where most cells have transitioned into a sessile state, we discover that motile cells can self-organize into two adjacent, centimeter-scale motile rings that surround the entire colony. Located at the outmost rim of the colony, the outer motile ring measures about ten microns in width; cells in this ring circle clockwise around the colony (viewing from above) with high polar order. On the other hand, the inner motile ring is several tens of microns in width; cells in this ring tend to swim in parallel to colony edge bidirectionally with nematic ordering in cells' moving directions. We demonstrate that the motile rings arise from spontaneous segregation of a homogeneous swimmer suspension that mimics a phase separation; the process is mediated by intercellular interactions and by a shear-induced depletion that concentrates bacteria in sheared regions. The remarkable self-organization of colony-scale motile rings is present in colonies of *Proteus mirabilis*, *Escherichia coli*, and *Bacillus subtilis*, suggesting that the phenomenon is conserved among bacteria species with peritrichous flagella. As a result of this self-organization, cells in the outer motile ring drive fluid flows in the inner motile ring to circulate counterclockwise (CCW) around the colony at a constant peak speed of ~30 μm s$^{-1}$, providing a stable and high-speed avenue for directed material transport at the macroscopic scale. Our findings reveal a unique form of colony-scale self-organization and active transport in bacterial colonies, which may shape the population structure and material distribution of bacterial communities in widespread environments.

## Results

**Self-organization of colony-scale motile rings.** Our organism of choice was *P. mirabilis*, a Gram-negative rod-shaped bacterium well known for its swarming behavior on hard agar surfaces[39,40]. *P. mirabilis* is widely distributed in the natural environment and as a human pathogen with clinical importance, it is often associated with urinary tract infections[41]. When grown on soft agar plates in humid environment (~85% relative humidity) for ~24 h (Methods; Supplementary Table 1), *P. mirabilis* colonies have not initiated swarming yet and most cells have become sessile, although a sub-population of motile planktonic cells (ranging from 2 to 10 μm in length) are present. Such sessile colonies of *P. mirabilis* provide a model system for studying the behavior and physiological function of motile sub-populations. We discovered that, surprisingly, motile cells in such colonies self-organize into two adjacent motile rings that surround the entire colony (Fig. 1a–d; Supplementary Movie 1). The outer motile ring is located at the outmost rim of the colony; it measures ~10 μm in width and ~1 μm in height. Cells in this ring are well aligned with each other, with the average orientation making an angle of ~30° with colony edge, presumably due to steric repulsion between densely packed cells[42]; they circle exclusively clockwise (CW) around the colony (viewed from above; observed in >100 *P. mirabilis* colonies on tens of agar plates) at a uniform, constant speed (mean: 27.8 μm s$^{-1}$, S.D.: 2.7 μm s$^{-1}$, $N = 10$), i.e., their collective motion is polarly ordered. On the other hand, the inner motile ring is ~20–40 μm in width and ~2–3 μm in height; it is bounded by the outer ring and the sessile part of the colony, and cells in this ring tend to swim in parallel to colony edge bidirectionally with nematic order in cells' moving directions (Fig. 1e, Supplementary Fig. 1; Supplementary Movie 2). Nonetheless, the motion of cells in the inner ring shows a weak CCW bias [defined as $N_{CCW}V_{CCW}/N_{CW}V_{CW}$, where $N_{CCW}$ (or $N_{CW}$) and $V_{CCW}$ (or $V_{CW}$) denote the number and the mean speed of CCW (or CW) moving cells, respectively] ranging from ~1.1 to ~1.2, and the inner motile ring collectively circles CCW around the colony at a mean speed of ~1 μm s$^{-1}$ (Fig. 1d). Our phenomenon is distinct from the self-organized vortex reported in microscale circular confinement[31,32], which did not display chirality bias and nematic order in cells' moving directions. Note that hereinafter in the paper CW and CCW refer to the sense of chirality with respect to the entire colony viewed from above, unless otherwise indicated.

To characterize the dynamics of motile ring development, we measured the collective speed of bacteria at colony edge over a time course of 7 h starting from ~16 h after colony inoculation. At the early stage of colony development, cells adapt to the surface environment, extract water from the substrate, and become able to move on agar surface. At this stage the motion of cells displayed certain degree of ordering (as shown by the non-zero mean tangential speed of ~4 μm s$^{-1}$ prior to time = 0 min in Fig. 1f). This stage lasts for several hours until ~20 h after colony

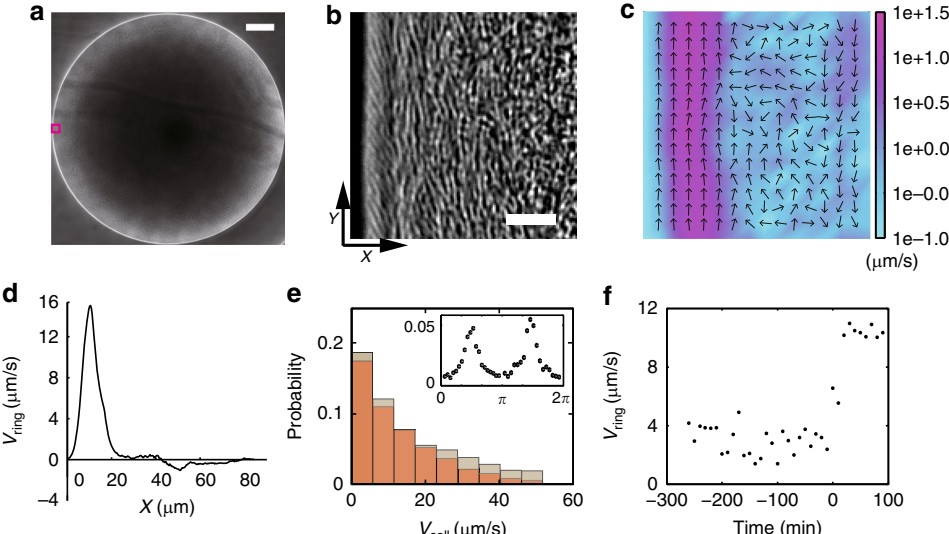

**Fig. 1** Self-organization of two adjacent colony-scale motile rings at the edge of a *P. mirabilis* colony. **a** Phase contrast image of *P. mirabilis* colony. Scale bar, 0.5 mm. **b** Enlarged view of the area enclosed by magenta box in panel A. Scale bar 10 μm. Also see Supplementary Movie 1. **c** Time-averaged collective velocity field of cells in the region of panel **b** computed by optical flow analysis ("Methods") based on phase contrast images. The collective velocity field was averaged over a duration of 10 s. Arrows represent velocity direction, and colormap represents velocity magnitude in log scale (with the color bar provided to the right, in μm s$^{-1}$). **d** The mean tangential speed of collective cellular motion (based on optical flow data) plotted against the distance from colony edge. Positive value of speed indicates motion along CW direction, i.e., along $+Y$ axis of the coordinate system specified in panel **b**, and $X = 0$ is set at the position of colony edge. **e** Speed distribution of individual bacteria at the inner motile ring moving in CCW direction (gray) and in CW direction (brown). Cell speed was computed by tracking fluorescent cells seeded into the colony ("Methods") and located near the center of inner motile ring where the magnitude of mean tangential speed was larger than 1/3 of the maximum CCW velocity magnitude in panel **d**. Inset: Probability distribution of velocity direction, with $3/2\pi$ and $1/2\pi$ corresponding to $-Y$ and $+Y$ directions, respectively. See Supplementary Fig. 1 and Supplementary Movie 2. **f** Dynamics of motile ring emergence during colony growth. The mean tangential speed of bacteria (computed by optical flow analysis with phase contrast images) in the 10-μm-wide outmost rim of colonies (i.e., the region of outer motile ring) is plotted against time. Time = 0 min is chosen at the onset of collective motion with high polar order and it corresponds to ~20 h colony growth

inoculation, at which time a highly organized outer motile ring very rapidly emerges at the colony edge. The full development is completed within 10 min, as shown by the sharp transition of mean tangential speed at $T = 0$ min in Fig. 1f. Since the motility and the density of cells at colony edge is similar before and after this transition point, we suggest that the physicochemical conditions of the colony, presumably water content (Supplementary Table 1) and surface tension, may just reached an appropriate state at the transition point permitting the formation of a thin wetting film ~1–2 μm in thickness at the colony edge that can support 2D motion of cells. Once emerged, the motile rings remain stable for many hours until the initiation of swarming. Environmental humidity is key to the formation of motile rings of *Proteus mirabilis*. Motile rings cannot be observed if the relative humidity is below 47% and they appear earlier on plates with higher agar concentrations (Supplementary Table 1).

To see whether this remarkable self-organization phenomenon was due to any physiological properties specific to *P. mirabilis*, we examined colonies of another two model flagellated bacteria, *E. coli* and *B. subtilis*. *E. coli* colonies were grown on LB agar plates that do not support swarm expansion ("Methods"), and we observed the same self-organization of motile rings described above (Supplementary Fig. 2 and Supplementary Movie 3), albeit less robust against environmental perturbations. *B. subtilis* displays vigorous swarming due to the secretion of surfactin, a lipoprotein having potent surfactant activities[43]. Nonetheless, *B. subtilis* colonies prior to the onset to swarm expansion (~3-h post inoculation; "Methods") displayed self-organization of motile rings as well (Supplementary Fig. 2 and Supplementary Movie 4). These findings suggest that the phenomenon is conserved among bacteria species with peritrichous flagella. We note that smooth-

swimming mutants of *E. coli* and *B. subtilis* do not display the self-organization of motile rings, at least under our experimental conditions. Smooth-swimming cells are not able to switch flagellar rotation direction autonomously. Consequently they tend to get stuck to each other during collisions and form jammed clusters. Indeed, jammed clusters frequently formed at the edge of such colonies (Supplementary Movie 5). Due to these jammed clusters, the orientation of cells at the colony edge remained disordered (Supplementary Fig. 3, panel A), and unidirectional motion could not develop there, despite a weak CW bias in the average speed of cells (Supplementary Fig. 3, panel B).

**Motile ring self-organization arises from physical interactions.** Next we sought to understand the self-organization process of motile rings. A recent study reported that concentrated swimming bacteria confined in 1D microfluidic channels can display unidirectional collective motion due to flagella-driven fluid flows generated by cells at channel edge[44]. The geometry in our phenomenon is similar to a 1-D channel, since the two motile rings are bounded by colony edge and the sessile part of the colony. In contrast to the phenomenon reported earlier[44], cells in the inner motile ring only display weakly directed collective motion (Fig. 1), suggesting a different mechanism at work. Nonetheless, we decided to examine whether channel-like confinement is necessary for motile ring self-organization. We collected motile cells from swarming colonies of *P. mirabilis*, re-suspended them in a medium containing surfactant Tween 20, and deposited them at appropriate densities on agar surface, forming a non-spreading liquid drop with motile cells alone that resembles a colony (referred to as an artificial colony; "Methods"). We found that motile cells in the artificial colony spontaneously separated into

three regions with different types of order, mimicking a phase separation: a dense and polarly ordered layer at the edge undergoing CW collective motion (~10 μm in width), a dense layer with nematic order in cells' moving directions just adjacent to the polar-order layer (~30 μm in width), and a low-density disordered phase in the bulk where cells swim randomly. The two densely-packed ordered layers resemble the colony-scale motile rings observed in naturally grown colonies (Fig. 2a–c, Supplementary Fig. 1; Supplementary Movies 6, 7). Our results suggest that channel-like confinement is not required for motile ring self-organization in colonies. The results also exclude the possibility that the sessile part of the colony may have orchestrated the collective motion in both motile rings through chemical signaling. We note that the speed distribution of cells in the nematic layer is Gaussian like (Fig. 2d), which is different from the speed distribution in the inner motile ring of colonies (Fig. 1e); this is because cells in the inner motile ring of colonies occasionally collide with sessile cells lying underneath and get stuck transiently, thus contributing to a large number of low-speed traces.

We further used suspension drops of *P. mirabilis* to characterize the dynamics of motile ring self-organization. We varied cell speed by tuning environmental temperature ("Methods") and found that the width of inner motile ring increases linearly with cell speed at the outer motile ring (Fig. 2e). This result suggests that physical interaction between cells, mediated by either steric repulsion or hydrodynamic forces or both, controls the formation of inner motile ring. We also tuned the cell density in suspension drops and found that the order of cellular motion in the 10-μm-wide outmost rim of the drop depends on cell density (Fig. 2f): At low densities, cellular motion in this rim does not display polar order, despite a weak CW bias (mean tangential speed ~2 μm s$^{-1}$); beyond a critical cell density (cell occupation ratio ~0.4), cellular motion experiences a sharp transition to maximal polar order (mean tangential speed ~12 μm s$^{-1}$), and a highly ordered CW motile ring emerges. This result demonstrates that the emergence of a highly ordered outer motile ring is mediated by intercellular interactions. The results presented in Fig. 1f and Fig. 2f reveals the two essential requirements of forming a highly ordered outer motile ring: (1) Appropriate physicochemical conditions of the colony edge that support 2D motion of cells; (2) Sufficiently high cell density that allows for intensive cell-cell interactions. As suggested earlier, the first requirement was not satisfied in naturally developed *P. mirabilis* colonies prior to $T = 0$ min in Fig. 1f, although the second requirement had already been met. By contrast, the first requirement was always satisfied in suspension drops of *P. mirabilis* with the help of exogenously supplied surfactant. We concluded that the self-organization of motile rings near the edge of the colony is a collective effect arising from physical interactions between swimmers under 2D confinement.

**CW bias is a result of cell-substrate hydrodynamic interaction**. To understand the origin of CW bias of collective motion in the outer motile ring, we examined the motion pattern of individual cells at low densities. *P. mirabilis* colony edge was diluted by adding external liquid, creating a liquid drop dispersed with isolated cells that mimicked the fluid environment of colonies ("Methods"). We found that cells colliding with the edge of such a diluted colony tended to turn CW with respect to the contact point about an axis normal to the underlying substrate and subsequently move to the right along the edge (Fig. 3; Supplementary Movie 8). In fact, cells moved exclusively CW after collision with the diluted colony edge for colliding angles between 0 and 1.75 rad (Fig. 3e). As expected, isolated cells of both *E. coli* and *B. subtilis* displayed similar CW bias of reorientation around

contact point (Supplementary Movie 9, 10). Interestingly, by imaging fluorescently labeled flagellar filaments of GFP-tagged *E. coli* cells[45], we found that the flagellar bundle of cells colliding with the edge of a suspension drop did not change its orientation with respect to the cell body throughout the turning process (Supplementary Movie 9; "Methods"). Presumably the CW bias of cell reorientation upon contact with colony edge gives rise to the exclusively CW bias of the collective motion in the outer motile ring of colonies.

To understand the mechanism of CW turning around contact point upon collision, we measured the time dependence of cell orientation and found that cells rotated at a constant angular speed $\omega$ during CW turning around contact point (for *P. mirabilis*, $\omega = 8.3$ (3.3) rad s$^{-1}$, mean (SD), $N = 56$; for *E. coli*, $\omega = 2.1$ (0.9) rad s$^{-1}$, mean (SD), $N = 15$). This constant angular speed of reorientation suggests the balance of torque about the cell pole in contact with drop edge. We build a simplified swimmer model to understand this torque balance, as illustrated in Fig. 3f:

$$\sum_i G_i = 0 \tag{1}$$

$$G_{\text{drive}} = G_{\text{drag}} \tag{2}$$

In our model, the driving torque $G_{\text{drive}}$ arises in the system because the bacterium rotates its flagella bundle CCW around flagellar axis (when looked from behind the cell body) next to the edge of the drop, where the bottom surface (agar) induces more friction than the top air-liquid interface[46]. $G_{\text{drive}}$ depends on the height of flagellar bundle above bottom surface. As the body with flagellar bundle reorients, the driving torque balances a resisting drag torque $G_{\text{drag}}$ due to rotation of the whole cell in the viscous fluid. Assuming that the flagella bundle is rotating about cell body long axis with angular speed $\Omega$, then the torque balance at low Reynolds number gives:

$$G_{\text{drive}} = D_1 \Omega, \tag{3}$$

$$G_{\text{drag}} = D_2 \omega, \tag{4}$$

$$\omega = \frac{D_1}{D_2} \Omega, \tag{5}$$

where $D_1$ is the constant reorientation coefficient relating the driving torque to the rotation rate $\Omega$ of the flagella bundle, and the constant coefficient $D_2$ relates the drag torque to the angular speed $\omega$ of cell body rotation about the cell pole in contact with drop edge. We derived $D_1$ and $D_2$ based on slender body theory[47,48] and resistive force theory[49], respectively ("Methods"), which allowed us to compute $\omega$ as a function of the flagellar bundle distance to the bottom surface $h$. Our simplified model is more appropriate for *E. coli* than for *P. mirabilis*, since *P. mirabilis* has 2–3 fold higher flagellar density on cell surface than *E. coli* and its flagellar filaments may not form a single bundle as modeled here[12]. Therefore we chose to verify our model with *E. coli*. Using the reorientation angular speed $\omega \approx 2.1$ rad s$^{-1}$ measured in our experiment as well as other known parameters of *E. coli*, we found the flagellar bundle distance to substrate as $h \approx 0.4$ μm ("Methods" and Supplementary Fig. 4), i.e., approximately half of cell body width. This result is reasonable, since cells are supposed to sit just above the substrate surface at the edge of liquid drop.

**Colony-scale directed flows are present in the inner motile ring**. Fluid pumping due to flagellar rotation of cells in the outer motile ring may generate strong unidirectional fluid flows, as suggested by previous studies[50,51]. To examine this idea, we

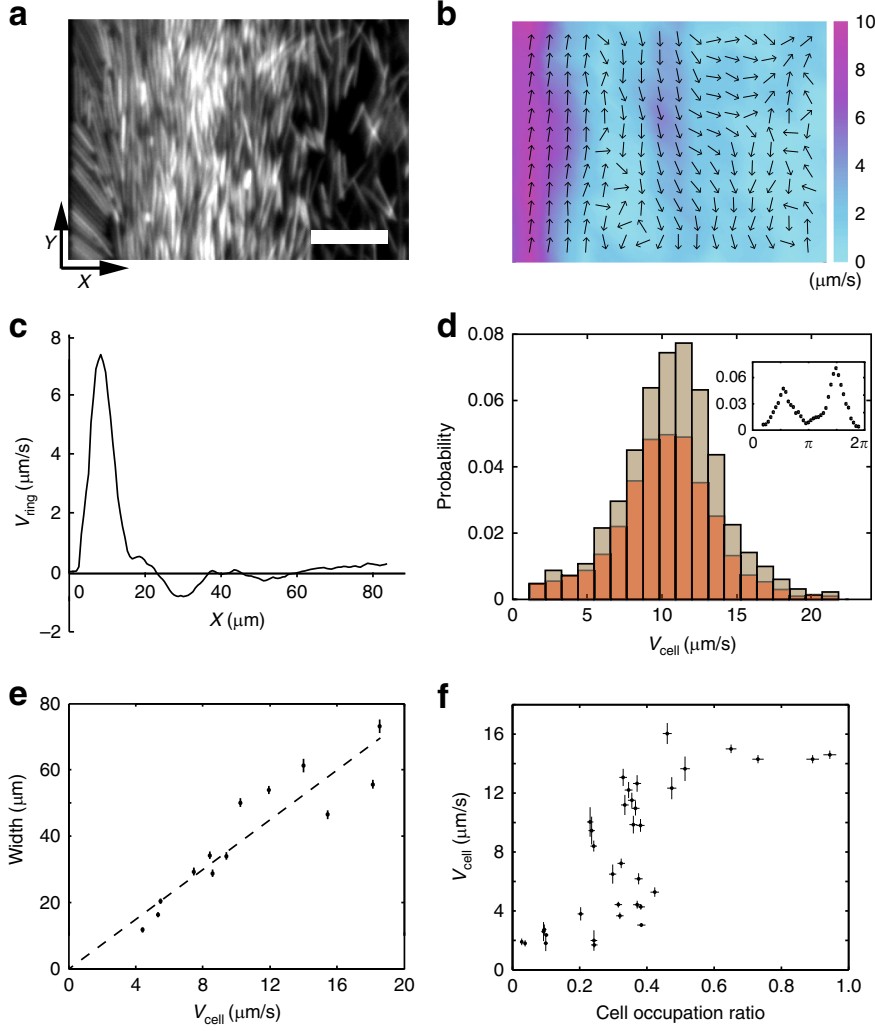

**Fig. 2** Self-organization of two adjacent motile rings at the edge of a *P. mirabilis* suspension drop (artificial colony). **a** Fluorescent image of the drop edge ("Methods"). Scale bar, 10 μm. See Supplementary Movie 6 (phase contrast) and Supplementary Movie 7 (fluorescence). **b** Time-averaged collective velocity field of cells in the region of panel **a** computed by optical flow analysis ("Methods") based on phase contrast images. The collective velocity field was averaged over a duration of 10 s. Arrows represent velocity direction, and colormap represents velocity magnitude (with the color bar provided to the right, in μm s$^{-1}$). **c** The mean tangential speed of collective cellular motion (based on optical flow data) plotted against the distance from suspension-drop edge. Following the coordinate system specified in panel **a**, positive value of speed indicates motion along $+Y$ axis, i.e., along CW direction viewed from above the suspension drop, and $X = 0$ is set at the position of drop edge. **d** Speed distribution of individual bacteria obtained by single-cell tracking at the inner motile ring where the magnitude of mean tangential speed was larger than 1/3 of the maximum CCW velocity magnitude in panel **c**. Brown and orange columns represent statistics for cells moving in CCW and CW direction, respectively. Inset: Distribution of velocity direction of individual bacteria; $3/2\pi$ and $1/2\pi$ correspond to $-Y$ and $+Y$ directions, respectively. Also see Supplementary Fig. 1. **e** The width of inner motile ring plotted against the mean tangential speed of collective motion at the outer motile ring. Dashed line is a linear fit with $R^2 = 0.84$. Error bars indicate standard error of the mean ($n = 1200$ measurements in three biologically independent samples). **f** The mean tangential speed of cells plotted against cell occupation ratio in the 10-μm-wide outmost rim of the suspension drop (corresponding to the region of outer motile ring in naturally developed colonies). Horizontal error bars indicate standard error of the mean ($n = 300$ measurements in three biologically independent samples); vertical error bars indicate standard deviation ($n = 10$ cells)

visualized fluid flows in *P. mirabilis* colonies by adding 0.1-μm fluorescent microspheres (Life Technologies, Cat. No. F8820) as tracers. We found that tracers were transported in a CCW manner around the entire colony at a peak constant speed of ~30 μm s$^{-1}$ (Fig. 4a–d; Supplementary Movie 11), along a narrow channel of width ~20–30 μm that coincides with the inner motile ring. A similar stream of fluid flows was found with a much smaller tracer, fluorescently labelled dextran of a size ~10 nm (FITC–Dextran, mol. wt. 2000 kDa; Sigma, FD2000S) that can move freely in between cells (Supplementary Fig. 5). The peak speed and the spatial range of fluid flows we uncovered are

comparable to those of the flagella-driven flows measured near individual cells whose motion were restricted in quasi-2D liquid films[50,51]. The CCW directed flows in the inner motile ring cause a shear rate up to ~10 s$^{-1}$ in the plane perpendicular to substrate surface; this shear rate falls within the range of intermediate shear rates (~9–12 s$^{-1}$) for slight downstream motion bias of flagellated bacteria[52], which may account for the weak CCW bias (i.e., downstream bias) of collective motion in the inner motile ring. Our flow measurement revealed a stable, high-speed avenue for directed material transport in bacterial colonies at the macroscopic scale. In addition, we also found that microspheres and

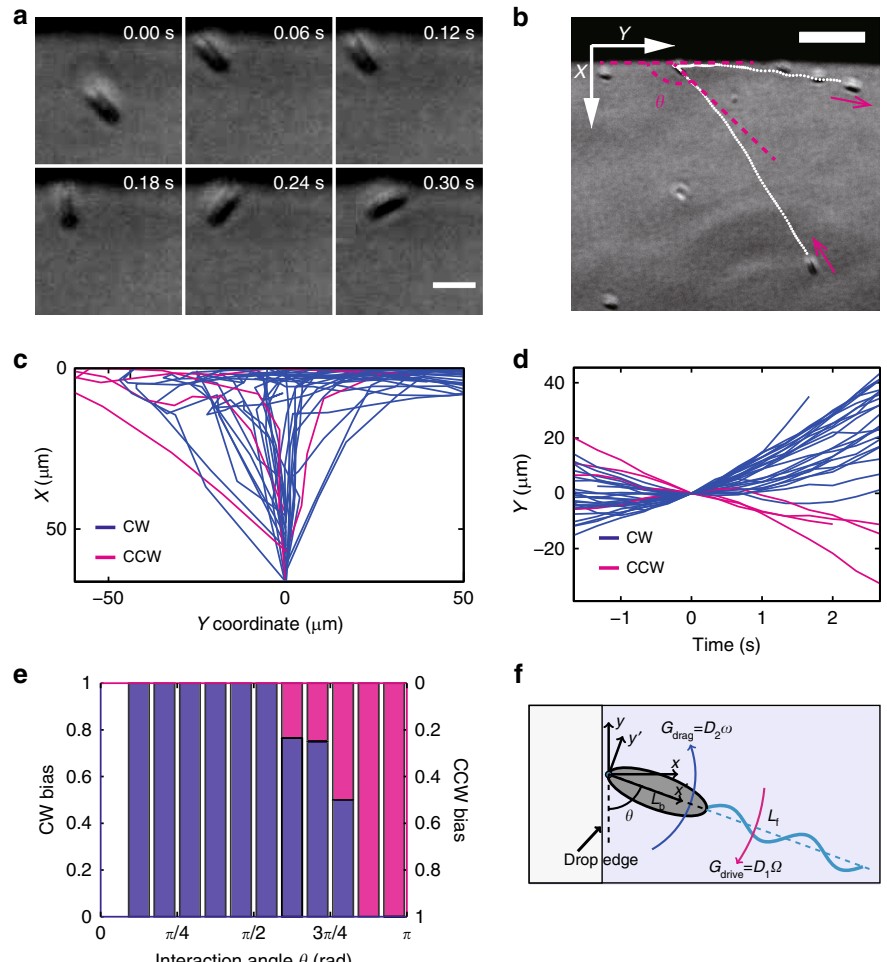

**Fig. 3** Single-cell motion pattern of *P. mirabilis* near diluted colony edge. **a** A sequence of phase contrast images recording the CW turning of an isolated cell colliding with the diluted colony edge ("Methods"). The dark area in upper portion of images is virgin agar. Scale bar, 5 μm. **b** Trajectory of the cell in panel **a** generated by overlaying a series of phase contrast images. Magenta arrows by the side of trajectory (white dotted line) indicate instantaneous moving direction of the cell. See Supplementary Movie 8. $\theta$ denotes interaction angle, defined as the angle between the diluted colony edge and the cell orientation just before collision. Scale bar, 20 μm. **c** Representative trajectories ($N = 31$) of cells that collided with diluted colony edge. Following the coordinate system specified in panel **b**, $X = 0$ μm is set at the edge of liquid drops. The starting point of all trajectories are aligned and set as $Y = 0$ μm and $X = 55$ μm. Blue (or magenta) lines represent the trajectory of cells that turned CW (or CCW) around the contact point during collision. **d** Tangential position of cells (whose trajectories were shown in panel **e**) plotted against time. The collision positions of all trajectories are aligned and set as $Y = 0$ μm, and $T = 0$ is set at the time of collision. Positive slope at $T > 0$ indicates that the cell moved in CW sense around the colony after collision. **e** Probability distribution of cells' motion bias after collision with diluted colony edge plotted against interaction angle (Blue column: CW bias; Magenta column: CCW bias). **f** Model of a swimming bacterium in contact with liquid drop edge (top view). The torque driving reorientation $G_{drive}$ and the drag torque $G_{drag}$ balance, which leads to rotation at a constant angular speed $\omega$ about the $x = 0$; $y = 0$ point (green dot). The flagellar bundle is rotating at an angular speed $\Omega$. The length of cell body and flagellar bundle is $L_b$ and $L_f$, respectively. Also see "Methods"

FITC–Dextran get transported into the sessile, interior part of the colony along some crack-like conduits (Fig. 4e–h; Supplementary Movie 12; Supplementary Fig. 5). The conduits typically formed at ~15–20 h after inoculation and became fully developed within 1 h (Supplementary Fig 6, panel A); they remained stable for many hours until the colony started to swarm, and then they gradually disappeared within ~3 h (Supplementary Fig. 6, panel B). The conduits have a mean radial extension of ~65 μm, with the longest ones up to ~200 μm (Supplementary Fig. 7). The inwards moving flows along the conduits are presumably driven by flagellar rotation of motile cells confined along the boundary of these conduits, and the flows may increase nutrient availability for the interior part of the colony.

To further verify that the observed CCW directed flows in the inner motile ring are driven by flagellar rotation of cells in outer motile ring, we used patterned illumination modulated by a digital mirror device (Andor Mosiac; "Methods") to optically deactivate the flagellar motility of cells in outer motile ring, while measuring how the transport of fluorescent microspheres in inner motile ring was affected. We took advantage of the photosensitizing effect of membrane stain FM 4–64 (Life Technologies, Cat. No. T13320; "Methods") for motility deactivation; upon excitation, FM4–64 disrupts flagellar motility, most likely due to the release of reactive oxygen species[53,54]. When cell motility was deactivated in a selected area of outer motile ring, we found rapid accumulation of microspheres in the region of inner motile ring just adjacent to the deactivation area (Fig. 5a–g) and the disappearance of CCW bias of collective motion in this region (Fig. 5h–i), indicating that CCW directed fluid flows were substantially halted in this region. Due to the substantial decrease

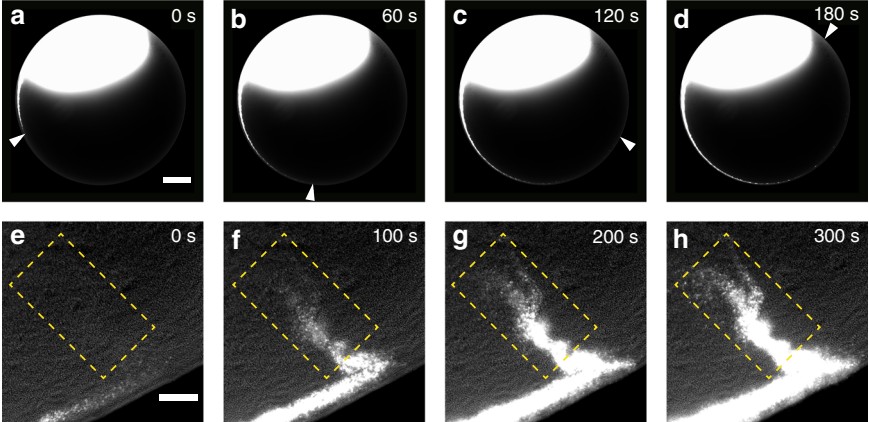

**Fig. 4** Long-range, colony-scale directed transport in a *P. mirabilis* colony. **a–d** Image sequence showing the directed transport of 0.1-μm fluorescent microspheres in the inner motile ring around colony edge ("Methods"). Fluorescent images of microspheres were overlaid with phase contrast images of the colony (at low light intensity). Bright blob in the upper part of the images is where the fluorescent microspheres were deposited. Arrows indicate the position of microspheres transported for the longest distance at indicated times; these foremost microspheres are only visible when enlarging the figure or when viewing the associated Supplementary Movie 11. Scale bar, 500 μm. **e–h** Image sequence showing the directed transport of 0.1-μm fluorescent microspheres inwards along crack-like conduits present near colony edge. The conduits are inside the rectangular area showed in the images. Also see Supplementary Movie 12. Scale bar, 50 μm

of supply from upstream, microspheres became barely visible in the inner motile ring downstream of the motility-deactivation region (Fig. 5f).

**Shear-induced depletion facilitates inner motile ring formation.** Here we introduce a conceptual model to explain the formation of inner motile ring. The results presented above provide strong evidence that the CCW directed flows in inner motile ring are driven by flagellar motility of cells in the outer motile ring. The directed fluid flows can cause shear-induced depletion to aid the formation of inner motile ring; shear-induced depletion is a phenomenon recently reported in bacterial suspensions that leads to the accumulation of swimmers in high-shear regions[55]. To examine this idea, we used fluorescent microspheres as tracers to map the flow speed profile generated by the CW outer motile ring of *P. mirabilis* suspension drops in the absence of the inner motile ring (thus avoiding the influence of cells in the inner motile ring on tracer motion, as microspheres tend to stick to cells) (Supplementary Fig. 8). We found that the region with horizontal shear rate coincided with the expected range of inner motile ring (ranging from ~10 to ~40 μm in X-axis of Supplementary Fig. 8). We further measured the radial profile of bacterial density in suspension drops (Supplementary Fig. 9). Near the transition region between the dilute phase and the inner motile ring (at ~40 micron from the edge), the surface cell density experienced a sharp increase when moving towards the edge. This increase of surface cell density cannot be attributed to the variation of the height of liquid film, since the fluid height in the dilute phase is comparable or greater than that in the inner motile ring; indeed, the increase of surface cell density reflects the increase of volume cell density and proves that cells accumulate towards the edge from the dilute phase. The region with cell accumulation (~10–40 micron from the edge) corresponds to the region of inner motile ring (Fig. 2a), and it coincides with the region with horizontal shear shown in Supplementary Fig. 8. These results (Supplementary Figs. 8, 9), together with the fact that the width of inner motile ring increases linearly with cell speed (Fig. 2e), supports the idea that shear-induced depletion helps attract cells to colony edge and contributes to the formation of inner motile ring in colonies.

## Discussion

Flagellated bacteria grown on solid substrates often develop into structured communities, in which most cells have transitioned into a non-motile state while a small population remains motile. In this paper, we investigated the behavior of motile cell population in sessile colonies of flagellated bacteria. Our major discovery is that motile cells in routinely cultured bacterial colonies can self-organize into two adjacent motile rings surrounding the entire colony via spontaneous segregation that mimics a phase separation. Cells in the outer motile ring circle CW around the colony (viewing from above) with high polar order, while cells in the inner ring swim in parallel to colony edge bi-directionally with nematic order in cells' moving directions. The self-organization of motile cells is mediated by intercellular interactions and shear-induced depletion. Flagellar rotation of cells in the outer motile ring generates directed fluid flows in the inner motile ring that circulate CCW around the colony at a constant speed of ~30 μm s$^{-1}$. Overall, our findings reveal that motile sub-populations in sessile colonies of flagellated bacteria can self-organize in a remarkable and heretofore unnoticed manner.

The mechanism underlying motile-ring self-organization deserves further study in the context of flagellar hydrodynamics and active matter self-organization. Shear-induced depletion may account for the accumulation of cells that leads to the formation of inner motile ring. As the inner motile ring is densely packed with swimming cells, would the modification of flow field by these cells affect the width or the stability of the motile rings? We note that these cells are moving bi-directionally and therefore the flows they generated would cancel one another out; consequently the flows generated by the outer motile ring would be largely preserved, as demonstrated by the strong CCW flows present in the inner motile ring in colonies (Fig. 4). Nonetheless, the weak CCW bias of collective motion in the inner motile ring would complicate the issue. Moreover, hydrodynamic attraction between cells, similar to that responsible for cohesive swimming of cells in 2D confinement[56], may help to reduce the orientational noise and thereby stabilize the motile rings. On the other hand, accumulation of cells via the shear-depletion mechanism is not sufficient to account for the nematic order in cells' moving directions in the inner motile ring. Advection due to fluid flows generated by flagellated bacteria at confinement boundary was shown to

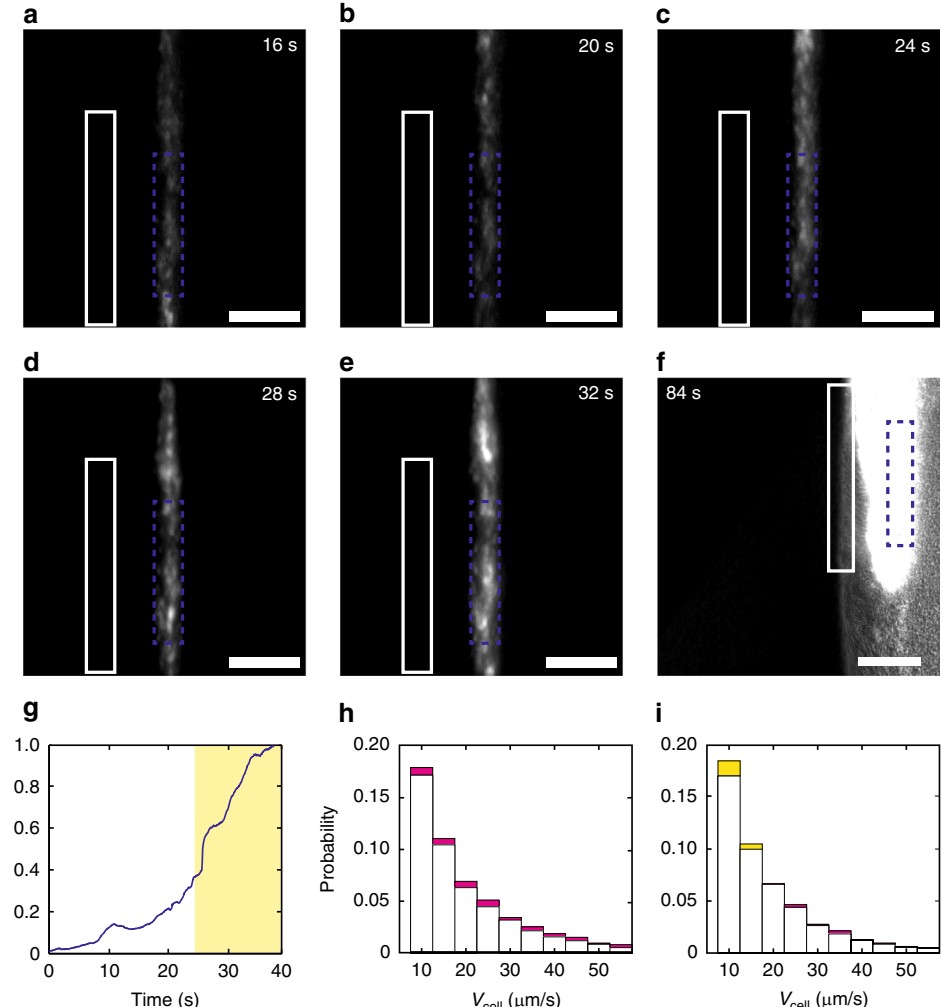

**Fig. 5** Optical deactivation of flagellar motility in the outer motile ring. A region of the outer motile ring (enclosed by solid white rectangles in panels **a–f** was chosen for motility deactivation with intense light ("Methods"). **a–e** Fluorescent image sequence of 0.1-μm microspheres in the same field of view before (**a–b**) and after (**c–e**) the deactivation of flagella motility. Motility-deactivation light at the selected region was turned on at time = 24 s (panel **c**). Scale bars, 50 μm. **f** Fluorescent image of 0.1-μm microspheres overlaid with the phase-contrast image of colony edge at time = 84 s. Microspheres were barely visible in the inner motile ring downstream of the motility-deactivation region. Scale bar, 50 μm. **g** Mean fluorescence intensity of microspheres averaged over a selected area of the inner motile ring (enclosed by blue rectangles in panels **a–f**) plotted against time. The shaded area starting at time = 24 s indicates the duration of motility-deactivation. Before motility deactivation (from 0 to 24 s), the slope of fluorescence intensity given by linear fit is $0.013 \pm 0.002$ a.u. s$^{-1}$ (±sign indicates fitting error; $R^2 = 0.94$); the slope increased to $0.041 \pm 0.002$ a.u. s$^{-1}$ ($R^2 = 0.95$) after motility deactivation. **h–i** Speed distribution of individual cells in the inner motile ring before (panel **h**) and after (panel **i**) motility deactivation, respectively. The speed distribution of cells moving in CCW or CW direction were represented by columns filled in magenta or yellow color, respectively, and both types of columns were plotted together in the same histogram with the overlapping portion filled with white color. CCW bias of cellular motion changed from ~1.19 to ~1.04 in this experiment

stabilize unidirectional collective motion (in which cells align their orientations nematically but move unidirectionally)[32], and nematically ordered collective motion (in which cells move bi-directionally in parallel with each other as seen in our phenom-enon) was not reported. In a recent study, computational mod-eling predicted that self-propelled polar particles interacting locally by nematic alignment could self-segregate into high-density bands with nematic order in particles' moving directions in the midst of low-density disordered regions[57]. It will be intriguing to examine the relevance of these model results to our findings. However, the model did not include hydrodynamic interaction between cells. The dynamics of motile rings may be better understood by modeling dense suspensions of swimmer that takes into account both hydrodynamic and steric interactions.

The unique mechanism of bacterial self-organization we uncovered here may be important to the physiology and stress response of bacterial communities in widespread environments. Self-segregation of motile subpopulations to the colony edge apparently reshapes the population structure in bacterial com-munities, which could facilitate the initiation of dispersal or range expansion (e.g., via swarming process). In addition, the self-organization of swimmers generates highly directed fluid flows, providing a stable high-speed avenue for long-range directed material transport in bacterial communities. Diffusion has been assumed to dominate material transport within bacterial com-munities[58–60], and only a few examples of directed transport have been found so far[51,61–63]. It was reported that *B. subtilis* colonies can make use of evaporation-driven fluid flows to aid colony-scale material transport;[62] however, the process is not driven by

autonomous cellular activities. Expanding swarms of flagellated bacteria can generate CW directed flows at colony edge at a similar magnitude ($\sim$10 µm s$^{-1}$)[51], but it is not stable due to constant swarm expansion (lasting for <1 min) and material transport can only persist for up to $\sim$500 µm; these time and length scales are two orders of magnitude smaller than that associated with the CCW flows in inner motile ring, which last for hours and persist for centimeters. Moreover, the long-range active transport we report here is enabled by bacterial self-organization, whereas cells at the edge of bacterial swarms do not display stable ordering of orientations or moving directions despite the presence of transient jets and vortices lasting for a fraction of a second. The unique means of long-range active transport in bacterial colonies we uncovered here is reminiscent of mucus flow driven by motile cilia of epithelial cells in various organ systems, such as the respiratory tract and oviduct[64]. We speculate that this long-range active transport may have profound effect on the physiology of bacterial communities. Bacteria communities in natural and clinical settings normally grow in anisotropic, structured environments with heterogeneous nutrient or chemical distribution. In such environments, bacterial communities do not have rotational symmetry and the self-organization of motile subpopulation may occur in various locations not limited to the colony edge; thus the long-range directed flows enabled by motile-cell self-organization may efficiently redistribute slowly-diffusing substances or molecules with high binding affinity to cell surface within bacterial communities. For example, high-molecular-weight polymeric metabolites[65] and membrane vesicles that encapsulate nutrients or signaling molecules[66] could travel for 1 cm in $\sim$5 min via the long-range directed flows, which is much more efficient than diffusion process alone (either passive diffusion driven by thermal energy or active diffusion driven by active forces in a bath of motile cells[61,67,68]). In fact, the 0.1-µm fluorescent microspheres we used to track the flows are comparable to the size of membrane vesicles secreted by many bacterial species, and thus the motion of microspheres should closely resemble the transport of membrane vesicles in situ in colonies. On the other hand, we wonder whether the long-range directed flows could be exploited to perturb the behavior of bacterial communities; for example, drugs and chemical effectors loaded in vesicles could be supplied locally and transported by the flows to manipulate distant parts of a bacterial colony.

## Methods

**Bacterial strains**. The following strains were used: wildtype *P. mirabilis* BB2000, and a fluorescent *P. mirabilis* KAG108 (BB2000 background with constitutive expression of Green Fluorescent Protein (GFP);[69] from Karine Gibbs, Harvard University, Cambridge, MA); *E. coli* HCB1737 (a derivative of *E.coli* AW405 with wildtype flagellar motility but with FliC S219C mutation for fluorescent labeling; from Howard Berg, Harvard University, Cambridge, MA); *E. coli* HCB1737 GFP (HCB1737 transformed with pAM06-tet plasmid with constitutive GFP expression[70] from Arnab Mukherjee and Charles M. Schroeder, University of Illinois at Urbana-Champaign); wildtype *B. subtilis* 3610 and a smooth-swimming mutant *B. subtilis* DK2178, which is a *cheB*-deleted derivative of *B. subtilis* 3610 (from Daniel B. Kearns, Indiana University at Bloomington). Single-colony isolates were grown overnight ($\sim$13–14 h) with shaking in LB medium (1% Bacto tryptone, 0.5% yeast extract, 0.5% NaCl) at 30 °C to stationary phase. For *E. coli* HCB1737 GFP, kanamycin (50 µg per mL) was added to the growth medium for maintaining the plasmid.

**Preparation of agar plates for colony growth**. LB agar (Difco Bacto agar at specified concentrations infused with 1% Bacto tryptone, 0.5% yeast extract, 0.5% NaCl) was autoclaved and stored at room temperature. Before use, the agar was melted in a microwave oven, cooled to $\sim$60 °C, and pipetted in 10-mL aliquots into 90-mm polystyrene petri plates. For optical deactivation of flagellar motility, the membrane stain FM 4–64 (Life Technologies, Cat. No. T13320) was dissolved in deionized water and added at a final concentration of 1.0 µg per mL to the liquefied agar before pipetting. The plates were swirled gently to ensure surface flatness, and then cooled for 10 min without a lid inside a large Plexiglas box. Drops of 1 µL overnight bacterial cultures were inoculated onto the LB agar plates. The inoculated

plates were dried for another 10 min without a lid inside the Plexiglas box. The plates were then covered and incubated at 30 °C and $\sim$85% relative humidity in an incubator with water tray for specified durations of time (see below). To vary the environmental humidity for colony growth, the plates were incubated at 30 °C in a custom-built incubator in which different kinds of saturated salt solutions (NaBr: 47.4%RH; KI: 55.0%RH; NaCl: 60.4%RH) or DI water ($\sim$97% RH) were supplemented, or incubated in a regular incubator without water tray ($\sim$42% RH) at 30 °C.

**Phase contrast and fluorescence imaging**. Imaging was performed on a motorized inverted microscope (Nikon TI-E). The following objectives were used in different experiments for both phase contrast and fluorescence imaging: Nikon CFI Plan Fluor DL4X, N.A. 0.13, W.D. 16.4 mm; Nikon CFI Achromat DL 10×, N.A. 0.25, W.D. 7.0 mm; Nikon CFI Super Plan Fluor ELWD ADM 20XC, N.A. 0.45, W.D. 8.2–6.9 mm; and Nikon CFI Super Plan Fluor ELWD ADM 40XC, N.A. 0.60, W.D. 3.6–2.8 mm. Fluorescence imaging was performed in epifluorescence using filter sets specified below, with the excitation light provided by a mercury pre-centered fiber illuminator (Nikon Intensilight). All recordings were made with a sCMOS camera (Andor ZYLA 4.2 PLUS USB 3.0) at full frame size (2048 × 2048 pixels) using the software NIS-Elements AR (Nikon). In all experiments the petri dishes were covered with lid to prevent evaporation and air convection, and the sample temperature was maintained at 30 °C using a custom-built temperature control system installed on microscope stage, unless otherwise stated.

**Bacterial self-organization at the edge of sessile colonies**. To observe self-organization at the edge of sessile colonies (Fig. 1 and Supplementary Fig. 2), *P. mirabilis* BB2000 or *E. coli* HCB1737 colonies were grown on 0.6% LB agar plates at $\sim$85% relative humidity (as described above) for 24 h, and *B.subtilis* colonies were grown on 2.0% LB agar plates $\sim$85% relative humidity for 2 h. Swarming of *P. mirabilis* occurs under our experimental conditions following the emergence of motile rings (>24 h after inoculation, with large variability of the exact initiation time of swarming). Observations were made in phase contrast with the 40× dark phase objective (Nikon CFI Super Plan Fluor ELWD ADM 40XC, N.A. 0.60, W.D. 3.6–2.8 mm) and were recorded at 30fps. For single-cell tracking at the edge of *P. mirabilis* colonies, overnight culture of the GFP-tagged *P. mirabilis* KAG108 was mixed with wildtype *P. mirabilis* at 1:2000 before inoculating the agar plates. *P. mirabilis* KAG108 cells were imaged with an FITC single band filter set (excitation 482/35 nm, emission 536/40 nm; dichroic: 506 nm; FITC-3540C-000, Semrock Inc.) and were recorded at 30 fps for 2 min. To monitor the development of motile rings and crack-like conduits during colony growth, the edge of *P. mirabilis* colonies was recorded at 30 fps for 10 s duration every 20 min with the 20× dark phase objective (Nikon CFI Super Plan Fluor ELWD ADM 20XC, N.A. 0.45, W.D. 8.2–6.9 mm).

**Bacterial self-organization at the edge of cell suspension drops**. To observe self-organization at the edge of cell suspension drops (artificial colonies) (Fig. 2), overnight culture of the GFP-tagged *P. mirabilis* KAG108 was either used alone (Fig. 2a, e, f, Supplementary Movies 6, 7) or mixed with wildtype *P. mirabilis* BB2000 at 1:50 (Fig. 2b–d), and 1-µL drops of the pure culture or the mixture were inoculated on 0.6% LB agar plates. The plates were incubated at 30 °C and $\sim$85% relative humidity for 36 h; at this time the colony had started to expand. Cells were collected from the leading edge of the colony with motility buffer (0.01 M potassium phosphate, 0.067 M NaCl, 10$^{-4}$ M EDTA, pH 7.0). The collected suspension was washed by mobility buffer for three times and resuspended in motility buffer to reach appropriate cell densities. Tween 20 (Sigma-Aldrich, Cat. No. P7949) was then added to the cell suspension at a final concentration of 0.002% (wt/wt). 1-µL drops of this suspension were deposited onto 2.0% LB agar plates supplemented with 0.002% (wt/wt) Tween 20. Bacterial self-organization at the edge of such suspension drops was observed in phase contrast and in fluorescence (via the FITC single band filter set described above) with the 40× dark phase objective. Images were recorded at 30 fps. To study the velocity dependence of the width of inner motile ring, we made use of the fact that the swimming speed of flagellated bacteria depends on temperature[71], and we varied cell speed in *P. mirabilis* suspension drops by changing environmental temperature from 15 to 37 °C with a temperature-controlled water bath.

**Single-cell motion pattern at the edge of liquid drops**. For *P. mirabilis* BB2000, 1 µL motility buffer was deposited near the edge of a *P. mirabilis* colony. The diluted colony edge formed a liquid drop dispersed with isolated cells. The motion pattern of isolated cells near the diluted colony edge was observed in phase contrast with the 20× dark phase objective and was recorded at 30 fps for 10 min (Fig. 3). For *B. subtilis*, overnight culture was diluted with motility buffer 10$^{-4}$, and 1-µL drops of the diluted suspension were deposited onto 2.0% LB agar plates. The motion pattern of isolated *B. subtilis* cells near the edge of the suspension drop was observed in phase contrast with the 20× dark phase objective and was recorded at 30 fps for 10 min (Supplementary Movie 10). For *E. coli* (HCB1737 GFP), flagellar filaments were fluorescently labeled prior to observation of single-cell motion pattern, following procedures established in ref. [45] and modified in ref. [56]. Briefly, 0.1 µL overnight culture of *E. coli* (HCB1737 GFP) was diluted 10$^{-4}$ to a volume of 1 mL in a glass test tube. Labeling dye stock solution (Alexa Flour 546 C$_5$-

maleimide, 5 mg per ml in DMSO; Life Technologies) was added to the cell suspension in test tube at a final concentration of 25 µg per ml. Labeling was allowed to proceed for 7 min in a shaker (30 °C and 180 rpm). When labeling was completed, the cell suspension was transferred to 1.5 ml Eppendorf tubes, brought to a volume of 10 mL with mobility buffer, and washed free of unreacted dyes by centrifugation and resuspension for three times ($10,000 \times g$ for 4 min in the first wash, $2000 \times g$ for 10 min in the second and third washes). After washing, cells were resuspended in mobility buffer to 1 mL, and 1-µl drops of this cell suspension were deposited onto 2.0% LB agar plates supplemented with 0.002% (wt/wt) Tween 20. The motion pattern of cell body and flagellar filaments of isolated *E. coli* cells near the edge of the suspension drop was imaged in fluorescence with the 20× dark phase objective, via an EGFP/mCherry dual band filter set (excitation: 471/38 nm and 571/34 nm, emission: 522/42 nm and 634/62 nm, dichroic: 522/52 nm and 636/84 nm; Part No. 59022, Chroma), and was recorded at 30 fps (Supplementary Movie 9).

**Visualization of fluid flows in colonies and suspension drops**. *P. mirabilis* BB2000 colonies were grown on 0.6% LB agar plates as described above for 24 h. To visualize fluid flows (Fig. 4, Supplementary Figs. 5 and 8), fluorescent microspheres (Fluo-Spheres, carboxylate-modified, 0.1 µm diameter; Life Technologies Cat. No. F8820) or fluorescently labeled dextran (FITC–Dextran, mol. wt. 2000 kDa; Sigma, FD2000S) were deposited near the colony edge. Fluorescence of microspheres was excited via a Cy3/TRITC single band filter set (excitation 535/50 nm, emission 610/75 nm, dichroic: 565 nm; Part No. 41007, Chroma), and FITC–Dextran was excited via the FITC single band filter set. Phase contrast images of colony edge (low-intensity illumination from above the sample provided by the built-in halogen lamp of the microscope) and fluorescent images of microspheres were taken with the 4× phase-contrast objective (Nikon CFI Plan Fluor DL4X, N.A. 0.13, W.D. 16.4 mm) and recorded simultaneously at 5 fps (Supplementary Movie 11). The directed transport of fluorescent microspheres inwards along crack-like conduits near colony edge was observed with the 10× objective (Nikon CFI Achromat DL 10X, N.A. 0.25, W.D. 7.0 mm), and was recorded at 5 fps simultaneously in phase contrast (illumination from above the sample provided by the built-in halogen lamp of the microscope) and in fluorescence (illumination provided by Nikon Intensilight via the Cy3/TRITC single band filter set) (Supplementary Movie 12). The transport of FITC–Dextran near colony edge was observed with the 10× objective and recorded at 5 fps in fluorescence (illumination provided by Nikon Intensilight via the FITC single band filter set). To measure the flow speed profile generated by CW outer motile ring (Supplementary Fig. 8), *P. mirabilis* BB2000 suspension drops were deposited onto 2.0% LB agar as described above, and fluorescent microspheres was deposited near the suspension drop edge. Fluorescence of microspheres was excited via the Cy3/TRITC single band filter set and recorded with the 20× objective at 30 fps.

**Optical deactivation of motility via patterned illumination**. *P. mirabilis* BB2000 colonies were grown on 0.6% LB agar plates supplemented with FM 4–64 (final concentration 1.0 µg per mL) as described above for 24 h. Fluorescent microspheres were deposited near the colony edge (described above) and got transported along the colony edge. A selected region of the outer motile ring (length: 150 µm; width: 20 µm) was chosen for motility deactivation. A beam of intense green-orange light provided by an LED illuminator (Thorlabs 530 nm Mounted High-Power LED L3, item No. M530L3, installed on X-Cite XLED1; Excelitas Technologies Corp) was modulated by a digital mirror device (Andor Mosaic) to form a spatially defined illumination pattern, and the modulated light passed through the 40× objective via the Cy3/TRITC filter set (described above) to excite FM4–64 in the selected region for motility deactivation. Meanwhile, the entire field of view was illuminated at low light intensity for imaging in phase contrast (illumination from above the sample provided by the built-in halogen lamp of the microscope) and/or in fluorescence (illumination provided by Nikon Intensilight via the Cy3/TRITC filter set). Recordings were made at 30 fps.

**Image processing and data analysis**. Images were processed using the open source Fiji (ImageJ) software (http://fiji.sc/Fiji) and custom-written programs in MATLAB R2016b (The MathWorks; Natick, Massachusetts, United States). The velocity field of cells' collective motion was obtained by performing optical flow analysis based on microscopy movies using the built-in functions of MATLAB. Prior to computing the optical flow fields, the images were first smoothed to reduce noise by convolution with a Gaussian kernel of standard deviation 1. The optical flow field for any two consecutive video frames was computed using the Horn-Schunck algorithm[72] (maximum iteration number, 128; smoothness parameter, 1) and then smoothed by local averaging. The grid size of the optical flow field was 1 pixel × 1 pixel and the initial value of optical flow vectors was set to zero. The obtained optical flow fields were used to compute the collective speed profile in Figs. 1d and 2c. To visualize the collective velocity field (Figs. 1c and 2b), the optical flow fields were coarsened to a grid size of 5.2 µm × 5.2 µm. The results were insensitive to different parameters of smoothing. For single-cell and single-microsphere tracking, bacteria and microsphere trajectories were obtained by a custom-written particle-tracking program in Matlab (The MathWorks, Inc) based on phase-contrast or fluorescence movies. Single-cell speed calculation was done in

different ways for the outer motile ring and for the inner motile ring, respectively. In the outer motile ring, cells move at an almost uniform speed, so we randomly chose 10 cells there and computed their mean speeds over ~10 s. The speed of cells in the inner motile ring has a broad distribution, so we tracked ~1000 cells in ~5-min movies and divided their trajectories into ~10,000 segments, each segment with a duration of 1 s. The mean speed of these 1-s segments were computed to yield the velocity statistics of cells. The velocity statistics of microspheres were obtained in a similar way. To study the cell density dependence of collective motion in outer motile ring, the cell occupation ratio at the 10-µm-wide outmost rim of suspension drops was defined as the area occupied by cells divided by total area of the rim. To determine the width of inner motile ring in *P. mirabilis* suspension drops, GFP-tagged *P. mirabilis* (KAG108) was used alone and imaged in fluorescence (as described above); the width of inner motile ring was taken as the region with fluorescence intensity greater than half of the maximum fluorescence at the edge of the suspension drop.

**Model of cell reorientation upon collision with colony edge**. The driving reorientation torque $G_{drive}$ is numerically estimated using the Slender Body Theory (SBT)[47,48] for the helix rotating above the no-slip wall ignoring the top air/fluid Interface. The SBT model takes into account full hydrodynamic interactions with the surface (except for lubrication when the helix is all but touching the surface). Firstly the force distribution $f$ on the stationary helix rotating next to the no-slip wall is calculated. From this force distribution we calculate the instantaneous reorientation torque about the tip of the cell ($x = 0$; $y = 0$ point in Fig. 3f):

$$G_{drive}(t) = \int_0^{L_f} (L_b + x')f_{y'}(t)\sec\Psi\,dx', \tan\Psi = \frac{2\pi R}{P} \quad (6)$$

where $f_{y'}\,(t, h, r, x')$ is the force density in the $y'$ direction which is driving the reorientation (Fig. 3f), $h$ the height above the wall of the helical axis, $r$ a radius of flagellar bundle, $L_f$ the length of the flagellum along its axis, $L_b$ the length of the cell body, $x'$ the length parameter $x' \in [0, L_f]$, $R$ the radius of flagellar helix, and $P$ the pitch of flagellar helix. The contribution due to the force component $f_x$ will be small because of the geometry. The instantaneous torque is averaged over one period $T$ of the flagellar rotation:

$$G_{drive} = 1/T \int_0^T G_{drive}(t)\,dt \quad (7)$$

The parameters for the helix are: the pitch $P = 2.3$ µm[73], radius $R = 0.3$ µm[73], the axial length $L_f = 7.3 \pm 2.4$ µm[73], the length of the cell body $L_b = 1.9$ µm[74], radius of the bundle $r = 24$ nm (double radius of a single flagellum)[75], frequency of rotation $1/T = \Omega/2\pi = 110$ Hz[73]. The driving torque as a function of height is computed as shown in Supplementary Fig. 4. The closer the flagellar bundle is to the bottom wall, the larger reorientation torque is created.

We use the Resitive Force Theory (RFT)[49] to estimate the drag torque $G_{drag}$ due to the rotation of the body with flagellar bundle around the tip of the cell. The majority of the drag torque comes from the flagellar bundle because it is much longer than the body and the flagellar bundle is further away from the rotation point. Therefore we approximate the cell-flagella system as a slender rod of length $L = L_b + L_f$ (which is a sum of the body length and flagellum length) and of radius $r$ (i.e., the radius of flagellar bundle). The force per unit length $\mathbf{f}$ on a slender straight filament moving with velocity $\mathbf{u}$ is given by:

$$\mathbf{f} = -(\xi_{\perp}\mathbf{nn} + \xi_{\parallel}\mathbf{tt}) \cdot \mathbf{u} \quad (8)$$

where $\mathbf{t}$ is the vector along the axis of filament and $\mathbf{n}$ perpendicular to it. In this case the body-flagella system is rotating about the tip of the cell. The point which is $l$ distance away from the tip is moving at the linear speed $\omega l$. The drag torque is given by:[76]

$$G_{drag} = -\int_0^L (\mathbf{f} \cdot \mathbf{n})l\,dl = \int_0^L (\xi_{\perp}\omega l)l\,dl = \xi_{\perp}\omega\frac{L^3}{3}$$

$$\xi_{\perp} = \frac{4\pi\mu}{1/2 + \ln(0.18P\sec\Psi/r)}, \tan\Psi = \frac{2\pi R}{P} \quad (9)$$

which means that the reorientation angular speed is

$$\omega = \frac{3G_{drive}}{\xi_{\perp}L^3} \quad (10)$$

We then use the result from the SBT for the driving torque $G_{drive}$ to calculate the reorientation angular speed $\omega$ as a function of $h$, as shown in Supplementary Fig. 4.

**Reporting summary**. Further information on research design is available in the Nature Research Reporting Summary linked to this article.

## Data availability

The data supporting the findings of this study are included within the paper and its Supplementary Information.

## Code availability

The custom codes used in this study are available from the corresponding author upon request.

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

## Acknowledgements

We thank Howard C. Berg, Karine Gibbs, Daniel B. Kearns, Arnab Mukherjee, and Charles M. Schroeder for their kind gifts of bacterial strains, and Howard C. Berg for helpful comments. The work of H.X. and Y.W. was supported by the National Natural Science Foundation of China (NSFC 21473152; to Y.W.) and by the Research Grants Council of Hong Kong SAR (RGC Ref. No. GRF 14322316&14301915, CUHK Direct Grants 4053230&4053310; to Y.W.). This project has also received funding from the European Research Council (ERC) under the European Union's Horizon 2020 research and innovation programme (grant agreement 682754 to EL).

## Author contributions

H.X. performed the experiments, analyzed and interpreted the data. J.D., D.D., and E.L. development the mathematical model. Y.W. discovered the phenomena, designed the study, performed the initial experiments, analyzed and interpreted the data. Y.W. wrote the paper, with input from other authors.

## Additional information

**Competing interests:** The authors declare no competing interests.

