## [Peer Review File · Nature Communications]

Reviewers' comments:

Reviewer #1 (Remarks to the Author):

Description of the work:

In this manuscript, the authors report the existence of two counter-rotating rings of motile bacteria surrounding sessile colonies of *Proteus mirabilis*, *Bacillus subtilis* and *Escherichia coli* growing on solid agar Petri dishes. By studying suspensions of motile bacteria within liquid droplet deposited on agar, they showed that those rings form spontaneously at the edge of the liquid droplet. Finally, they injected fluorescent microspheres at the edge of the colony in order to observe their dispersal. Microspheres are advected in the motile rings around the periphery of the colony until they reach a "crack" that facilitates their access towards the center of the colony.

General opinion:

This paper has a good potential to attract readers in the fields of active matter and collective phenomena. Even if the physics that underlies phase separation in bacterial suspensions has already been well characterized in defined geometries, the paper investigates how much biologically relevant this phenomenon is for material transport during colony growth. To make their point, key experiments addressing the formation dynamics of motile rings and cracks during colony growth are missing.

Major comments:

1. *Proteus mirabilis* is a strain that is well known for its ability to swarm on agar Petri dishes. Could the authors explain in more details why they do not observe swarming in their experimental configuration?
2. Taken individually many observations reported in this article have been studied in well-defined geometries: a) Active bacterial suspensions in droplets have been extensively studied by the group of R. Goldstein (Wioland et al, PRL 2013; Lushi et al., PNAS 2014). These studies have reported the formation of two concentric rings of bacteria moving in opposite direction near the edge of the droplet. b) It has also been shown that when bacteria encounter a wall, they swim parallel to it (see Frymer et al., PNAS 1995; Lushi et al., PNAS 2014; Sipos et al., PRL 2015; Bianchi et al., Phys.Rev.X 2017). c) Finally, in channels, shear depletion mechanisms have been shown to concentrate bacteria in high shear regions (Rusconi et al., Nat. Phys. 2013). All these works (the list I provided is not exhaustive) deserve a deep description in the introduction.
3. The very nice aspect of this study is to reposition all the ingredients listed above in the more physiological and natural context of colony growth. But to strengthen the description they made, it would be great to address when the motile ring starts to form during colony growth (in terms of time relative to inoculation or colony size) and why bacteria make an angle of 30° with colony edge.
4. The authors present convincing evidence that isolated bacteria preferentially reorient clockwise at the edge of droplet containing a dilute suspension of bacteria. But, it is somehow surprising that in colony experiments the outer ring always rotates clockwise regardless of perturbations that may occur at the edge of the colony. Is it really the case if 10 independent replicates (10 colonies on distinct Petri dishes) are analyzed?
5. The authors have used 100nm microspheres as reporter for material transport inside the colony. Yet, the large size of these particles compared to the size of the gaps between bacteria in the dense sessile phase of the colony may restrict their motion to the inner ring, which is less dense. It is therefore important to study smaller particles, like fluorescein, in order to assess whether active transport dominates over diffusion only when particles are large or more generally for all nutrients diffusing in the environment.
6. If advection by the circular flow in the inner ring is important for allowing material to be transported at distance of the injection site, the presence of cracks is as much important for material transport towards the center of the colony. Hence, from a bacterial perspective, the strategy for optimizing nutrient uptake by the colony necessitates the presence of cracks. Could

the authors document and analyze in more detail crack formation during colony growth (using the same set of suggested experiments in point 3)? They could measure the probability of formation as a function of time or colony size, their distribution along the contour of the colony and their radial extension from the edge.

Minor comments:

- Line 32: "peritrichously flagellated" is redundant.
- Line 112: The way the speed is computed is not explicitly exposed. Is the value given here an average over 10 individual trajectories? How speed is computed from individual trajectories (average or max value)? In the supplementary information, the authors mention the tracking of 1000 trajectories. Could the authors clarify this point in the supplementary information?
- Line 116: Fig. 1D should be Fig. 1E,F.
- Line 121: Fig. 1D,E should be Fig. 1D.
- Line 148-150: Reference 34 does not provide the statistics of individual trajectories, so the comparison should be made with optical flow analysis and not individual trajectories.
- Line 149: Fig. 1 should be Fig. 1E,F.
- In Fig. 1E,F, it would be more instructive to plot the whole distribution of angles in the same plot (as in Fig. 2D).
- In Fig. 1C, the color scale is not appropriate for displaying speed variations over two orders of magnitude. Could the authors represent this plot in a log scale so that we could distinguish the inner ring and figure out the noise?
- Line 161: Fig. 2 should be Fig. 2A-C.
- Line 162: We already know from previous studies that channel confinement is not required for motile ring self-organization (see major comment 2).
- Line 169-171: Reference 35 does not mention the existence of a critical size over which vortex formation is suppressed.
- Fig. 2C: Could the authors explain why the average velocity of the outer ring in the dilute suspension is smaller than in the colony (because of a lower density)?
- Fig. 2E,F: Could the authors explain why the distribution of individual speeds is gaussian for the dilute suspension instead of exponential as observed in the colony?
- Line 320: References 34 and 35 report the existence of a chiral nematic phase.
- Line 333: I do not understand on which basis the authors make this statement.
- Line 340-356: This part of the discussion is a bit speculative.
- Line 439: There is no Fig. S2b in the supplementary information (Fig S2 has y-axes on both left and right axis but just one set of data points).

Reviewer #2 (Remarks to the Author):

I found assessing this article difficult. On the one hand, the authors seem to have discovered a novel property of some bacterial colonies: the existence of an outer boundary layer made of few cells which are rotating clockwise (CW) on average, and of a second, larger, boundary layer with nematic order. Discovering something new about such broadly-studied problems is always likely to interest many people.

On the other hand, neither the biological significance of this discovery nor its underlying physical mechanisms are thoroughly detailed. I find that the whole article contains a lot of untested hypothesis and claims, and that the whole presentation of the results is rather confusing. For

instance, the authors keep referring to the 'collective motions' of the cells, but the argument they invoke to explain the CW motion is a one-body effect. In physics, collective phenomena rely on the interactions between the agents, not on the existence of an external driving force (here, the drop edge).

For these reasons, in its present shape, I cannot recommend publishing this article in Nature Communication. A thorough rewriting of the paper, together with an important complementary work to characterize either the physics or the biology of these boundary layers would be needed to change my mind. That said, I find this discovery quite nice and I am thus disappointed with the content of my review, which I would have liked more positive. The authors can find a list below of the main points which prevented from supporting publications, beyond the one listed above.

The authors keep referring to 'phase separation'. From what they describe, however, there are no 'phases' apart from the sessile cells in this systems, since both layers have finite widths that do not scale with the colony size: they are simply boundary layers.

Along the same lines, the authors keep referring to these layers as 'large-scale' structures, but I have to disagree with that statement: they simply scale with the perimeter of the colony, as any boundary layer would.

Another terminology I find unconvincing is 'self-organization'. The authors are quite explicit that the outer layer is simply due to the boundary condition: this is not 'self' organized but, on the contrary, externally driven. What could have been described as a self-organization is the second layer, but the authors are very vague about its origin. First, the width of the first layer (and hence the beginning of the second layer) is not accounted for. Then, the connection to shear-induced depletion to explain the second layer is interesting but not tested. Furthermore, this hypothesis is questioned by the impact of the presence of the cells on the actual flow. Finally, the origin of the transition between the second layer and the bulk of the system is not explained. (This transition is actually barely described.)

Active particles are known to do surprising things close to boundaries and surfaces are known to bend the trajectory of swimmers, so that I side with the authors when they claim that the interactions between the cells and the drop edge in the vicinity of the underlying surface is probably responsible for the CW motion. I am, however, surprised by their argument about the hydrodynamic torques created by the flagellar bundle being responsible for this CW bias. Indeed, I remember having looked at bacteria colliding with (hard) walls and did not observe such a clear bias. I noticed that G_{drive} is decreasing as h increases on Figure S2, but I wonder at which distance this effect would become negligible. More importantly, when h is around $0.4\mu\text{m}$, I would expect the cells to actually touch the surface, so that the main origin of the torque would be the friction between the cell body and the gel: the cells would then reorient much like a rotating pen does when its tip is touching a piece of paper. Can the authors rule out this effect? Finally, I am not familiar with slender body theory: does it rely on far field hydrodynamics? If so, isn't this approximation violated when the cell is so close to the edge drop & to the gel surface? If the full hydrodynamic flow is accounted for, then it would be a good idea to indicate so for the uneducated reader (or referee).

Beyond these precise points, I found frustrating that the authors present a phenomenon, but do not discuss the conditions for its occurrence (i.e., the control parameters): would the presence of any circular boundary lead to this CW motion? Does it depend on the contact angle of the drop in which the bacteria live? What controls the widths of both layers? What happens precisely for the smooth swimmers case reported by the authors, which does not show such driven CW motion?

More minor points: I found the connection to active rods (lines 321-323) surprising since the dense lanes described there seem to be lacking here, and the order seems to have a completely different origin (alignment upon collision vs presence of an ordering boundary). I was also

surprised to read about the lack of self-organization of swarming cells (lines 347-348) since I thought that the swarming of *B. subtilis* was actually explained precisely by a complex self-organization (see, e.g., Hamze et al, *microbiology* 157:2456-246 (2011)).

Response to the comments of Reviewer #1:

We thank the reviewer for the encouraging comments and helpful suggestions. Here are our point-by-point responses:

Reviewer #1 (Remarks to the Author):

General opinion:

This paper has a good potential to attract readers in the fields of active matter and collective phenomena. Even if the physics that underlies phase separation in bacterial suspensions has already been well characterized in defined geometries, the paper investigates how much biologically relevant this phenomenon is for material transport during colony growth. To make their point, key experiments addressing the formation dynamics of motile rings and cracks during colony growth are missing.

Response: In this revised version we have made substantial efforts to elucidate the conditions, dynamics, and physical mechanisms of motile ring formation. The following is a list of major new results:

(1) We documented the timing and investigated the environmental determinants (agar concentration and humidity) of motile ring formation during *P. mirabilis* colony growth. We found that the onset of motile rings occurs at ~20 hr after colony inoculation, at which point a highly organized CW outer motile ring emerges at the colony edge very rapidly within 10 min. We also found that there exists a critical environmental humidity for the development of motile rings, below which the colonies initiate swarming within 5 hr after inoculation. Please see Fig. 1F and Table S1 in the manuscript; also see response to Major comments Point 1,3 below.

(2) We studied the self-organization dynamics of motile rings under constant environmental conditions using suspension drops of *P. mirabilis*. We found that the width of inner motile ring increases linearly with cell speed at the outer motile ring, and that the motion of cells in the outer motile ring experiences a sharp transition to maximal polar order beyond a critical cell density (surface packing ratio of ~0.4). The latter result shows that the emergence of highly ordered CW outer motile ring is a collective effect arising from cell-cell interactions, but not merely a result of cell-boundary interaction. Please see Fig. 2E&F in the manuscript; also see response to Major comments Point 3 below.

(3) To investigate whether shear-induced depletion could account for the accumulation of cells in the inner motile ring, we mapped the flow speed profile generated by the CW outer motile ring of *P. mirabilis* suspension drops in the absence of the inner motile ring. We found that the region with horizontal shear rate $> \sim 1 \text{ s}^{-1}$ coincided with the expected range of inner motile ring. This result, together with the fact that the width of inner motile ring increases linearly with cell speed (Fig. 2E), supports the idea that shear-induced depletion helps attract cells to colony edge and contributes to the formation of inner motile ring in colonies. Please see Fig. 2E and Fig. S7 in the manuscript.

(4) We characterized the dynamics of crack development in *P. mirabilis* colonies qualitatively. We found that the timing of crack formation is close to but a little earlier than the full development of motile rings. The cracks remained stable for many hours until the colony started to swarm, and then the cracks gradually disappeared within ~3 hr. Please see Fig. S5 and Fig. S6 in the manuscript; also see response to Major comments Point 6 below.

Major comments:

1. *Proteus mirabilis* is a strain that is well known for its ability to swarm on agar Petri dishes. Could the authors explain in more details why they do not observe swarming in their experimental configuration?

Response:

In fact, swarming does occur under our experimental conditions (0.6% LB agar in 85% relative humidity), and it follows the emergence of motile rings (>24 hr after inoculation, with large variability of the exact initiation time of swarming). Motile rings were probably overlooked in many earlier studies that focused on the swarming stage alone. Also the environmental humidity is key to the formation of motile rings of *Proteus mirabilis*. By varying environmental humidity and agar concentration when growing *P. mirabilis* colonies, we found that motile rings could not be observed below a critical relative humidity (somewhere between 47-55%); instead, the colonies initiated swarming within 5 hr after inoculation if the relative humidity was below 47% (Table R1). We speculate that a drier environment could increase viscosity within the colony and promote swarm-cell differentiation (Allison et al., 1993; see full reference below). On the other hand, agar concentration does not affect the results qualitatively but only changes the timing of motile ring formation. The results presented in Table R1 will be further discussed with regard to the onset of motile rings in Response to Major Comments Point 3. We have revised lines 118-119&160-162 and SI Methods to include these results.

Reference: Allison, C., Lai, H.-C., Gygi, D., & Hughes, C. (1993). Cell differentiation of *Proteus mirabilis* is initiated by glutamine, a specific chemoattractant for swarming cells. *Molecular Microbiology*, 8(1), 53–60.

A

The effect of LB agar concentration on the growth of *P. mirabilis* colonies with relative humidity 85%RH

	LB concentration (%)			
	0.6	1.0	1.5	2.0
Sessile colony with motile rings	Yes	Yes	Yes	Yes
Initiation of motile rings (hour)	19 ± 1	19 ± 1	17 ± 1	17.5 ± 0.5
Swarming	Yes	Yes	Yes	Yes
Initiation of swarming (hour)	> 24	> 24	> 24	> 24

B

The effect of relative humidity on the growth of *P. mirabilis* colonies with LB agar concentration 0.6%.

	Relative humidity (%RH)					
	97.0	85.0	60.4	55.0	47.4	42.0
Sessile colony with motile rings	Yes	Yes	Yes	Yes	n.a.	n.a.
Initiation of motile rings (hour)	19 ± 1	19 ± 1	15.5 ± 0.5	14.5 ± 0.5	n.a.	n.a.
Swarming	Yes	Yes	Yes	Yes	Yes	Yes
Initiation of swarming (hour)	> 24	> 24	> 24	> 24	4.5 ± 0.5	4.5 ± 0.5

Table R1. (A) The effect of LB agar concentration on the growth of *P. mirabilis* colonies cultured at 85% relative humidity. (B) The effect of relative humidity on the growth of *P. mirabilis* colonies cultured on 0.6% LB agar.

2. Taken individually many observations reported in this article have been studied in well-defined geometries: a) Active bacterial suspensions in droplets have been extensively studied by the group of R. Goldstein (Wioland et al, PRL 2013; Lushi et al., PNAS 2014). These studies have reported the formation of two concentric rings of bacteria moving in opposite direction near the edge of the droplet. b) It has also been shown that when bacteria encounter a wall, they swim parallel to it (see Frymer et al., PNAS 1995; Lushi et al., PNAS 2014; Sipos et al., PRL 2015; Bianchi et al., Phys.Rev.X 2017). c) Finally, in channels, shear depletion mechanisms have been shown to concentrate bacteria in high shear regions (Rusconi et al., Nat. Phys. 2013). All these works (the list I provided is not exhaustive) deserve a deep description in the introduction.

Response:

In previous version some of the listed references (Wioland et al, PRL 2013; Lushi et al., PNAS 2014; Rusconi et al., Nat. Phys. 2013) were discussed in Results section. Here we have

revised the last paragraph in Introduction (lines 93-108) as follows and clearly pointed out the connection of our findings to earlier studies on bacterial behavior in confinement:

“Indeed, our findings are relevant to the behavior of confined bacterial suspensions in more artificial settings. It was reported that confined bacterial suspension droplets of size less than ~100 micron^{32,33} can self-organize into vortices, in which two concentric, counter-rotating regions of bacteria are present near the edge of the droplet. Our results present the first observation of this type of bacterial self-organization in the context of naturally developed, millimeter-scale colonies; moreover, cells in our phenomenon display chirality bias and nematic order in cells’ moving directions that were not reported in previous research^{32,33}. Bacteria swimming near liquid-solid interfaces are well known to have the tendency moving in parallel to the wall due to hydrodynamic trapping³⁴⁻³⁸, but it was less clear how swimming bacteria behave at the gas-liquid-solid three phase interface, such as the edge of bacterial colonies, and our findings advance the understanding of this problem. Despite these connections to previous studies of bacterial swimming behavior in artificial settings, the mechanisms underlying the self-organization of swimmers into motile rings in bacterial colonies deserve further study in the context of flagellar hydrodynamics and active matter self-organization.”

The references included in these lines are as follows:

- 32 Wioland, H., Woodhouse, F. G., Dunkel, J., Kessler, J. O. & Goldstein, R. E. Confinement Stabilizes a Bacterial Suspension into a Spiral Vortex. *Physical Review Letters* **110**, 268102 (2013).
- 33 Lushi, E., Wioland, H. & Goldstein, R. E. Fluid flows created by swimming bacteria drive self-organization in confined suspensions. *Proceedings of the National Academy of Sciences* **111**, 9733-9738, doi:10.1073/pnas.1405698111 (2014).
- 34 Frymier, P. D., Ford, R. M., Berg, H. C. & Cummings, P. T. Three-dimensional tracking of motile bacteria near a solid planar surface. *Proceedings of the National Academy of Sciences* **92**, 6195-6199 (1995).
- 35 Berke, A. P., Turner, L., Berg, H. C. & Lauga, E. Hydrodynamic attraction of swimming microorganisms by surfaces. *Phys Rev Lett* **101**, 038102 (2008).
- 36 Molaei, M., Barry, M., Stocker, R. & Sheng, J. Failed Escape: Solid Surfaces Prevent Tumbling of Escherichia coli. *Physical Review Letters* **113**, 068103 (2014).
- 37 Sipos, O., Nagy, K., Di Leonardo, R. & Galajda, P. Hydrodynamic Trapping of Swimming Bacteria by Convex Walls. *Physical Review Letters* **114**, 258104, doi:10.1103/PhysRevLett.114.258104 (2015).
- 38 Bianchi, S., Saglimbeni, F. & Di Leonardo, R. Holographic Imaging Reveals the Mechanism of Wall Entrapment in Swimming Bacteria. *Physical Review X* **7**, 011010, doi:10.1103/PhysRevX.7.011010 (2017).

3. The very nice aspect of this study is to reposition all the ingredients listed above in the more physiological and natural context of colony growth. But to strengthen the description they made, it would great to address when the motile ring start to form during colony growth (in terms of time relative to inoculation or colony size) and why bacteria make an angle of 30° with colony edge.

Response:

(a). Regarding the first question (i.e. “when the motile ring start to form during colony growth”), we have carefully monitored the timing of motile ring emergence in naturally developed colonies. We measured the collective speed of bacteria at colony edge over a time course of 7 hours starting from ~16 hours after colony inoculation, via long-term time lapse microscopy. At the early stage of colony development, cells adapt to the surface environment, extract water from the substrate, and become able to move on agar surface. At this stage the motion of cells displayed certain degree of ordering (as shown by the non-zero mean tangential speed of ~4 micron/s prior to Time=0 min in Fig. R1), presumably due to the CW bias of cell reorientation when colliding with the edge of colony (Fig. 3 in main text). This stage lasts for several hours until ~20 hr after colony inoculation, at which time a highly organized outer motile ring very rapidly emerges at the colony edge: The full development is completed within 10 min, as shown by the sharp transition of mean tangential speed at T=0 min in Fig. R1. The results are incorporated in lines 145-162 and Fig. 1F of the revised manuscript.

The result presented in Fig. R1 suggests that either the physiology of cells in the colony or the physicochemical conditions of the colony just reached an appropriate state for motile ring emergence immediately prior to the transition point at T= 0 min. As cells already started moving many hours before this transition point, the physiological state of cells must be similar before and after the transition; so it is more likely that the physicochemical conditions of the colony, such as water content or surface tension, are playing the key role. These factors are difficult to manipulate in naturally developed colonies because the process of water extraction and the molecular details of biosurfactant synthesis are unclear (see the references Hennes et al. 2017 and Zhang et al. 2010 listed below). Nonetheless, when we prepare artificial colonies (i.e. suspension drops), we notice that appropriate amount of surfactant (Tween 20; ~0.002% wt/wt) has to be supplemented to the medium. This fact suggests that surface tension of the colony has to be sufficiently low, such that a thin wetting film ~1 micron in thickness can form at the colony edge to support 2D motion of cells. So we believe surface tension is a main determinant for motile rings to emerge in naturally developed colonies. On the other hand, water content of the colony may be an important factor for the timing of onset of motile rings as well. We varied agar concentration and environmental humidity when growing *P. mirabilis* colonies and found that motile rings appeared earlier on agar plates with higher agar concentrations, and appeared earlier in less humid environment (but the humidity cannot be lower than 47-55%, otherwise the colony swarms quickly) (see Table R1 in Response to Point 1). The environmental humidity may affect the water extraction process of the colony or the production of surfactant-like substances. However, the underlying mechanism is not clear to us and it is worth further study. This discussion is incorporated in lines 145-162 and Table S1 of the revised manuscript.

In addition, we used suspension drops of *P. mirabilis* to characterize the dynamics of motile ring self-organization under constant environmental conditions. We varied cell speed by tuning environmental temperature (from 15 °C to 37°C) and found that the width of inner motile ring increases linearly with cell speed at the outer motile ring (Fig. R2, panel A). This result suggests that physical interaction between cells, mediated by either steric repulsion or hydrodynamic forces or both, controls the formation of inner motile ring. We also tuned the cell density in suspension drops and found that the order of cellular motion in the 10- μ m-wide outmost rim of the drop depends on cell density (Fig. R2, panel B): At low densities, cellular motion in this rim does not display polar order, despite a weak CW bias (mean tangential speed ~2 μ m/s) due to the CW bias of cell reorientation upon contact with drop edge; beyond a critical cell density (cell occupation ratio ~0.4), cellular motion experiences a sharp transition to maximal polar order (mean tangential speed ~12 μ m/s), and a highly ordered CW motile ring emerges. This result demonstrates that the emergence of a highly ordered outer motile ring is a self-organized collective effect arising from cell-cell interactions; meanwhile, the CW chirality of

outer motile rings presumably results from the CW bias of cell reorientation upon contact with drop edge. These results are incorporated in lines 213-227 and Fig. 2E-F of the revised manuscript.

References:

1. Hennes, M., Tailleur, J., Charron, G., & Daerr, A. (2017). Active depinning of bacterial droplets: The collective surfing of *Bacillus subtilis*. *Proc. Natl. Acad. Sci. USA*. 114 (23): 5958-5963.
2. Zhang, R., Turner, L., & Berg, H. C. (2010). The upper surface of an *Escherichia coli* swarm is stationary. *Proc. Natl. Acad. Sci. USA*, 107, 288–290.

Fig. R1. Dynamics of motile ring emergence during colony growth. The mean tangential speed of bacteria (computed by optical flow analysis with phase contrast images) in the 10- μm -wide outmost rim of colonies (i.e. the region of outer motile ring) is plotted against time. Time = 0 min is chosen at the onset of collective motion with high polar order and it corresponds to ~ 20 hr colony growth.

Fig. R2. (A) The width of inner motile ring plotted against the mean tangential speed of collective motion at the outer motile ring. Cell velocity was varied by changing environmental temperature from 15 $^{\circ}\text{C}$ to 37 $^{\circ}\text{C}$ and was computed by optical flow analysis based on fluorescent images (see SI Methods). Dashed line is a linear fit with $R^2 = 0.84$. (B) The mean tangential speed of collective motion plotted against cell occupation ratio in the 10- μm -wide

outmost rim of the suspension drop (corresponding to the region of outer motile ring in naturally developed colonies). The mean speed of cells was computed by optical flow analysis base on fluorescent images. The cell occupation ratio was defined as the area occupied by cells divided by total area of the 10- μ m-wide rim.

(b). For the second question (i.e. “*why bacteria make an angle of 30° with colony edge*”), the angle is presumably due to steric repulsion between densely packed cells at the colony edge. This is supported by the simulations of Wensink & Löwen (Physical Review E 78: 031409, 2008), who showed that self-propelled rods with pure steric interactions do not align parallel to a confining wall but tilt at an angle to the wall. The lines 126-127 are revised accordingly in the manuscript.

4. The authors present convincing evidence that isolated bacteria preferentially reorient clockwise at the edge of droplet containing a dilute suspension of bacteria. But, it is somehow surprising that in colony experiments the outer ring always rotates clockwise regardless of perturbations that may occur at the edge of the colony. Is it really the case if 10 independent replicates (10 colonies on distinct Petri dishes) are analyzed?

Response:

Yes the chirality of outer motile ring in colonies is robust. We have observed >100 *P. mirabilis* colonies with motile rings on tens of agar plates (0.6% LB, 85% relative humidity), and the outer motile rings of all these colonies display clockwise rotation. This fact is supplemented in lines 128-129 of the revised manuscript.

5. The authors have used 100nm microsphere as reporter for material transport inside the colony. Yet, the large size of these particles compared to the size of the gaps between bacteria in the dense sessile phase of the colony may restrict their motion to the inner ring, which is less dense. It is therefore important to study smaller particles, like fluorescein, in order to assess whether active transport dominates over diffusion only when particles are large or more generally for all nutrients diffusing in the environment.

Response:

To verify the results of material transport obtained by 100 nm microspheres, we used fluorescently labelled dextran of a size ~10 nm (FITC–Dextran, mol. wt. 2000 kDa; Sigma, FD2000S) to track the fluid flows. Molecules of this size can move freely in between cells. We found that the dextran solution deposited near colony edge also got transported rapidly in a counterclockwise manner along the colony edge, and some were transported inwards along the crack-like conduits (Fig. R3). This result is incorporated in lines 294-296 and Fig. S4 of the revised manuscript. We further tested calcein, whose molecular weight is less than 1 kDa, but such small molecules diffuse rapidly away from the inner motile ring as the bulk solutions are being transported. So we propose that the directed flows would be most suitable for molecules with high binding affinity to cell surface and for slowly-diffusing substances (e.g. polymeric metabolites and membrane vesicles that encapsulate nutrients or signaling molecules), as pointed out in Discussion of main text.

Fig. R3. Long-range, colony-scale directed transport in a *P. mirabilis* colony revealed by FITC–Dextran. Bright blob in the left part of the images is the deposited dextran solution. Scale bar, 500 μm .

6. *If advection by the circular flow in the inner ring is important for allowing material to be transported at distance of the injection site, the presence of cracks is as much important for material transport towards the center of the colony. Hence, from a bacterial perspective, the strategy for optimizing nutrient uptake by the colony necessitates the presence of cracks. Could the authors document and analyze in more detail crack formation during colony growth (using the same set of suggested experiments in point 3)? They could measure the probability of formation as a function of time or colony size, their distribution along the contour of the colony and their radial extension from the edge.*

Response:

We monitored the crack formation process via long-term time lapse microscopy, in the same way as the characterization of motile ring formation dynamics in response to Point 3. We found that crack formation typically began at ~15-20 hr after inoculation and cracks became fully developed within 1 hour (Fig. R4, panel A). The timing of crack formation is close to but a little earlier than the full development of motile rings. The cracks remained stable for many hours until the colony started to swarm, and then the cracks gradually disappeared within ~3 hr (Fig. R4, panel B). The probability distribution of cracks along the colony edge appeared to be random, and the distribution of their radial extension from the edge approximately follows exponential decay with a mean of ~65 μm (Fig. R5). The longest cracks extended up to ~200 μm inside the colony. These results have been incorporated in lines 307-311 and Fig. S5-S6 of the revised manuscript.

We concur with the reviewer in the potential biological significance of the cracks. We do not have a clear idea on the physical or biological mechanism of crack formation. It could be due to mechanical stress resulting from cell growth or due to cell de-differentiation from sessile to motile state. We would like to investigate this phenomenon in future work and focus on reporting the discovery of motile rings in the current paper.

Fig. R4. Formation and disappearance of cracks near *P. mirabilis* colony edge. (A) Image sequence showing the time-averaged collective velocity field of cells during crack formation (computed by optical flow analysis). (B) Image sequence showing the time-averaged collective velocity field of cells during crack disappearance (computed by optical flow analysis). Note that, as the lifetime of cracks spans many hours and we found that the cracks are prone to environmental perturbation, crack formation (panel A) and disappearance (panel B) were imaged separately. Scale bars, 50 μm . T=0 in panel A and B corresponds to 15 hr and 48 hr after inoculation, respectively.

Fig. R5. Distribution of radial extension of cracks. Cracks were visualized by FITC–Dextran at 24 hr after inoculation (SI Methods of the manuscript). The data are based on 33 cracks visualized in 20 colonies, and only the cracks with a radial extension greater than 40 μm are considered.

Minor comments:

- Line 32: "peritrichously flagellated" is redundant.

Response: In this work we did not study polarly flagellated bacteria such as *Vibrio* species. It would be interesting to examine in future work whether our findings can be observed in those species as well.

- Line 112: The way the speed is computed is not explicitly exposed. Is the value given here an average over 10 individual trajectories? How speed is computed from individual trajectories (average or max value)? In the supplementary information, the authors mention the tracking of 1000 trajectories. Could the authors clarify this point in the supplementary information?

Response: We apologize for the confusion. Single-cell speed calculation was done in different ways for the outer motile ring and for the inner motile ring, respectively. In the outer motile ring, cells move at an almost uniform speed, so we randomly chose 10 cells there and computed their mean speeds over ~10 s. The speed of cells in the inner motile ring has a broad distribution, so we tracked ~1000 cells in ~5-min movies and divided their trajectories into ~10,000 segments, each segment with a duration of 1 s. The mean speed of these 1-s segments were computed to yield the speed distribution of cells. We have clarified this point in supplementary information SI Methods.

- Line 116: Fig. 1D should be Fig. 1E,F.

- Line 121: Fig. 1D,E should be Fig. 1D.

- Line 149: Fig. 1 should be Fig. 1E,F.

- In Fig. 1E,F, it would be more instructive to plot the whole distribution of angles in the same plot (as in Fig. 2D).

- In Fig. 1C, the color scale is not appropriate for displaying speed variations over two orders of magnitude. Could the authors represent this plot in a log scale so that we could distinguish the inner ring and figure out the noise?

Response: We apologize for these careless mistakes. The comments are addressed in the new version of Figure 1. Please note that Fig. 1 is updated with a new panel on the formation dynamics of motile rings during colony growth. The panels E&F in the previous version are now combined into a single panel (E).

- Line 148-150: Reference 34 does not provide the statistics of individual trajectories, so the comparison should be made with optical flow analysis and not individual trajectories.

Response: This sentence is revised as follows in lines 190-191: "In contrast to the phenomenon reported earlier (Wioland et al., 2016), cells in the inner motile ring only display weakly directed collective motion".

- Line 161: Fig. 2 should be Fig. 2A-C.

Response: This mistake is now corrected.

- Line 169-171: Reference 35 does not mention the existence of a critical size over which vortex formation is suppressed.

Response: As shown by the experiment and simulation results in Fig. 4 of Ref. 35 (Ref. 33 in revised version), the polar order of vortex formed by dense bacterial suspensions drops significantly when the diameter of confinement is beyond $\sim 70 \mu\text{m}$. Nonetheless, for dilute or semidilute suspensions, the authors of Ref. 35 (Ref. 33 in revised version) noted that the drop center was depleted in cells, thus leading to ordered states in the boundary layer even for confinement of larger sizes. So we deleted these lines to avoid confusion.

- Fig. 2C: Could the authors explain why the average velocity of the outer ring in the dilute suspension is smaller than in the colony (because of a lower density)?

Response: The speed difference is most likely due to cell length difference. As shown in SI Movies S7 and S8, the cell length in suspension drops ($8.6 \pm 2.5 \mu\text{m}$) was about 60% longer than that at colony edge ($5.5 \pm 1.0 \mu\text{m}$), because cells in suspension drops were collected from the leading edge of swarming colonies where cells had differentiated into swarmer state. Longer cells experience more drag, while their propulsive force may not increase in proportion with the increase of drag.

- Fig. 2E,F: Could the authors explain why the distribution of individual speeds is gaussian for the dilute suspension instead of exponential as observed in the colony?

Response: There are some sessile cells lying underneath the inner motile ring, which can be seen in SI Movie 1. Motile cells occasionally collide with these sessile cells and get stuck transiently, thus contributing a large number of low-speed traces during single-cell tracking. By contrast, all cells in the suspension drop are motile, so their speed distribution is Gaussian like. We have clarified this point in main text (lines 207-211).

- Line 162: We already know from previous studies that channel confinement is not required for motile ring self-organization (see major comment 2).

Response: Since the focus of this sentence is on motile ring formation in colonies, we revised the sentence as follows to be more specific: "Our results suggest that channel-like confinement is not required for motile ring self-organization in colonies."

- Line 320: References 34 and 35 report the existence of a chiral nematic phase.

Response: We checked Refs. 34 and 35 (now Refs. 44 and 33 in revised version) again more carefully. In fact, the nematic order in their contexts referred to the cell-body orientations but not cells' moving directions; these cells aligned their orientations nematically but moved *unidirectionally* with polar order. In our case, by "nematic order" we refer to the nematic ordering of cells' moving directions, *i.e.* cells moving *bi-directionally* and in parallel with each other. We have revised this line (now lines 384-385) as "...nematically ordered collective motion (in which cells move bi-directionally in parallel with each other as seen in our phenomenon) was not reported". We have also clarified this point by rephrasing "nematic order" to be "nematic order in cells' moving directions" or equivalent throughout the main text.

- Line 333: I do not understand on which basis the authors make this statement.

Response: To clarify the point we have revised the sentence as follows: “Self-segregation of motile subpopulations to the colony edge apparently reshapes the population structure in bacterial communities, which could facilitate the initiation of dispersal or range expansion (e.g. via swarming process).”

The previous version of this sentence is quoted below for reference:

“Phase separation of motile subpopulations apparently reshapes the population structure in bacterial communities, which could facilitate the initiation of dispersal or range expansion.”

- *Line 340-356: This part of the discussion is a bit speculative.*

Response: This part (now in lines 416-424) was meant to discuss the potential physiological benefit of the long-range active transport enabled by motile rings. We agree that the proposed ideas are speculative. As the focus of the current paper is on reporting the discovery of motile rings, we hope the discussion could intrigue readers to examine the physiological roles of colony-scale material transport enabled by motile rings in diverse contexts of bacterial communities.

- *Line 439: There is no Fig. S2b in the supplementary information (Fig S2 has y-axes on both left and right axis but just one set of data points).*

Response: This mistake has been corrected. The driving reorientation torque G_{drive} and the reorientation angular speed were plotted in separate panels in an earlier version of Fig. S2; later both panels were merged in the current form of Fig. S2 because G_{drive} and the reorientation angular speed are linearly related, but the citations to Fig. S2 in main text were not updated correctly.

Response to the comments of Reviewer #2:

We thank the Reviewer for the encouraging comments and helpful suggestions. Here are our point-by-point responses:

1. I found assessing this article difficult. On the one hand, the authors seem to have discovered a novel property of some bacterial colonies: the existence of an outer boundary layer made of few cells which are rotating clockwise (CW) on average, and of a second, larger, boundary layer with nematic order. Discovering something new about such broadly-studied problems is always likely to interest many people.

On the other hand, neither the biological significance of this discovery nor its underlying physical mechanisms are thoroughly detailed. I find that the whole article contains a lot of untested hypothesis and claims, and that the whole presentation of the results is rather confusing. For instance, the authors keep referring to the 'collective motions' of the cells, but the argument they invoke to explain the CW motion is a one-body effect. In physics, collective phenomena rely on the interactions between the agents, not on the existence of an external driving force (here, the drop edge).

For these reasons, in its present shape, I cannot recommend publishing this article in Nature Communication. A thorough rewriting of the paper, together with an important complementary work to characterize either the physics or the biology of these boundary layers would be needed to change my mind. That said, I find this discovery quite nice and I am thus disappointed with the content of my review, which I would have liked more positive. The authors can find a list below of the main points which prevented from supporting publications, beyond the one listed above.

Response: We apologize for any confusing statements in the previous manuscript. In this revised version we have made substantial efforts to elucidate the conditions, dynamics, and physical mechanisms of motile ring formation. The following is a list of major new results:

(1) We documented the timing and investigated the environmental determinants (agar concentration and humidity) of motile ring formation during *P. mirabilis* colony growth. We found that the onset of motile rings occurs at ~20 hr after colony inoculation, at which point a highly organized CW outer motile ring emerges at the colony edge very rapidly within 10 min. We also found that there exists a critical environmental humidity for the development of motile rings, below which the colonies initiate swarming within 5 hr after inoculation. Please see Fig. 1F and Table S1 in the manuscript; also see response to Point 6 below.

(2) We studied the self-organization dynamics of motile rings under constant environmental conditions using suspension drops of *P. mirabilis*. We found that the width of inner motile ring increases linearly with cell speed at the outer motile ring, and that the motion of cells in the outer motile ring experiences a sharp transition to maximal polar order beyond a critical cell density (surface packing ratio of ~0.4). The latter result shows that the emergence of highly ordered CW outer motile ring is a self-organized collective effect arising from cell-cell interactions, but not merely a result of cell-boundary interaction. Please see Fig. 2E&F in the manuscript; also see response to Point 4 below.

(3) To investigate whether shear-induced depletion could account for the accumulation of cells in the inner motile ring, we mapped the flow speed profile generated by the CW outer motile ring of *P. mirabilis* suspension drops in the absence of the inner motile ring. We found that the region with horizontal shear rate $> \sim 1 \text{ s}^{-1}$ coincided with the expected range of inner motile ring. This result, together with the fact that the width of inner motile ring increases linearly with cell speed at the outer motile ring (Fig. 2E), supports the idea that shear-induced depletion helps attract cells to colony edge and contributes to the formation of inner motile ring in colonies. Please see Fig. 2E and Fig. S7 in the manuscript; also see response to Point 4 and Point 6 below.

(4) We characterized the dynamics of crack development in *P. mirabilis* colonies qualitatively. We found that the timing of crack formation is close to but a little earlier than the full development of motile rings. The cracks remained stable for many hours until the colony started to swarm, and then the cracks gradually disappeared within ~ 3 hr. Please see Fig. S5 and Fig. S6 in the manuscript.

2. The authors keep referring to 'phase separation'. From what they describe, however, there are no 'phases' apart from the sessile cells in this systems, since both layers have finite widths that do not scale with the colony size: they are simply boundary layers.

Response: Although the coexistence between regions of different types of order in our phenomenon mimics a phase separation, we thank the reviewer for pointing out the more rigorous criterion for using this term. We have changed 'phase separation' into 'self-organization' or 'segregation', and changed 'phase' to 'layer' or 'region' when describing the phenomenon throughout the text (including the title).

3. Along the same lines, the authors keep referring to these layers as 'large-scale' structures, but I have to disagree with that statement: they simply scale with the perimeter of the colony, as any boundary layer would.

Response: To avoid confusion, we have changed the 'large-scale' to 'colony-scale' when referring to the motile rings, and to 'long-range' when referring to the material transport along the inner motile ring.

4. Another terminology I find unconvincing is 'self-organization'. The authors are quite explicit that the outer layer is simply due to the boundary condition: this is not 'self' organized but, on the contrary, externally driven. What could have been described as a self-organization is the second layer, but the authors are very vague about its origin. First, the width of the first layer (and hence the beginning of the second layer) is not accounted for. Then, the connection to shear-induced depletion to explain the second layer is interesting but not tested. Furthermore, this hypothesis is questioned by the impact of the presence of the cells on the actual flow. Finally, the origin of the transition between the second layer and the bulk of the system is not explained. (This transition is actually barely described.)

Response:

(a). To clarify the self-organization nature of the CW outer motile ring, we tuned the cell density in suspension drops of *P. mirabilis* and found that the order of cellular motion in the 10- μm -wide

outmost rim of the drop depends on cell density (Fig. R1). At low densities, cellular motion in this rim does not display polar order, despite a weak CW bias (mean tangential speed $\sim 2 \mu\text{m/s}$) due to the CW bias of cell reorientation upon contact with drop edge; beyond a critical cell density (cell occupation ratio ~ 0.4), cellular motion experiences a sharp transition to maximal polar order (mean tangential speed $\sim 12 \mu\text{m/s}$), and a highly ordered CW motile ring emerges. This result demonstrates that the emergence of a highly ordered outer motile ring is a self-organized collective effect arising from cell-cell interactions; meanwhile, the CW chirality of outer motile rings presumably results from the CW bias of cell reorientation upon contact with drop edge. This result is incorporated in lines 213-227 and Fig. 2F of the revised manuscript.

Fig. R1. The mean tangential speed of collective motion plotted against cell occupation ratio in the 10- μm -wide outmost rim of the suspension drop (corresponding to the region of outer motile ring in naturally developed colonies). The mean speed of cells was computed by optical flow analysis base on fluorescent images. The cell occupation ratio was defined as the area occupied by cells divided by total area of the 10- μm -wide rim.

(b). To further understand the self-organization dynamics of the inner motile ring, we varied cell speed in suspension drops of *P. mirabilis* by tuning environmental temperature (from 15 °C to 37°C) and found that the width of inner motile ring increases linearly with cell speed at the outer motile ring (Fig. R2). This result suggests that physical interaction between cells, mediated by either steric repulsion or hydrodynamic forces or both, controls the formation of inner motile ring. In addition, to investigate whether shear-induced depletion could account for the accumulation of cells in the inner motile ring, we used fluorescent microspheres to map the flow speed profile generated by the CW outer motile ring of *P. mirabilis* suspension drops in the absence of the inner motile ring (thus avoiding the influence of cells in the inner motile ring on tracer motion, as microspheres tend to stick to cells) (Fig. R3). We found that the region with horizontal shear rate $> \sim 1 \text{ s}^{-1}$ coincided with the expected range of inner motile ring (ranging from ~ 10 to $\sim 40 \mu\text{m}$ in X-axis of Fig. R3). This result, together with the fact that the width of inner motile ring increases linearly with cell speed, supports the idea that shear-induced depletion helps attract cells to colony edge and contributes to the formation of inner motile ring in colonies. These results have been incorporated into the paper (Fig. 2E and Fig. S7 in the manuscript).

The shear-depletion mechanism explains the width of the inner motile ring. The mechanism indicates that the inner motile ring transits to the bulk disordered phase where shear rate diminishes to below $\sim 1 \text{ s}^{-1}$ (i.e. beyond $\sim 40 \mu\text{m}$ in X-axis of Fig. R3). Note that the starting position of the inner motile ring is always at $\sim 10 \mu\text{m}$ from the edge, i.e. the CW outer motile ring consists of on average one to two layers of cells that are in direct contact with the colony edge; other cells trying to adhere to the CW outer motile ring would be affected by the CCW shear flows generated by the outer motile ring and thus could not move along in a stable manner.

The inner motile ring is densely packed with swimming cells. As the reviewer pointed out, it is possible that the impact of the presence of these cells could affect the width and stability of inner motile ring. However, these cells are moving bi-directionally and therefore the flows they generated would cancel one another out; consequently the flows generated by the outer motile ring would be largely preserved, as demonstrated by the strong CCW flows present in the inner motile ring in colonies. Nonetheless, the weak CCW bias of collective motion in the inner motile ring would complicate the issue. Moreover, hydrodynamic attraction between cells, similar to that responsible for cohesive swimming of cells in 2D confinement (Li et al., PRL 2017), may help to reduce the orientational noise and thereby stabilize the motile rings. These points are discussed in revised Discussion section (lines 370-379). A better understanding of the width of both motile rings as well as the stability of inner motile ring may require modeling of dense suspensions of swimmer that takes into account both hydrodynamic and steric interactions, which would be out of the scope of the current paper.

Fig. R2. The width of inner motile ring plotted against the mean tangential speed of collective motion at the outer motile ring. Cell velocity was varied by changing environmental temperature from 15 °C to 37°C and was computed by optical flow analysis based on fluorescent images (see SI Methods). Dashed line is a linear fit with $R^2 = 0.84$.

Fig. R3. Mean tangential flow speed plotted against the distance from the edge of a *P. mirabilis* suspension drop. Fluid flows were visualized by 0.1 μm diameter microspheres and the flow

speed was measured by tracking single microspheres. To avoid the influence of cells in the inner motile ring on microsphere motion (as microspheres tend to stick to cells), flow visualization was done when only the CW outer motile ring appeared at the edge and the inner motile ring had not developed yet. Positive value of speed indicates motion along CCW direction, and $X = 0$ is set at the position of colony edge.

5. Active particles are known to do surprising things close to boundaries and surfaces are known to bend the trajectory of swimmers, so that I side with the authors when they claim that the interactions between the cells and the drop edge in the vicinity of the underlying surface is probably responsible for the CW motion. I am, however, surprised by their argument about the hydrodynamic torques created by the flagellar bundle being responsible for this CW bias. Indeed, I remember having looked at bacteria colliding with (hard) walls and did not observe such a clear bias. I noticed that G_{drive} is decreasing as h increases on Figure S2, but I wonder at which distance this effect would become negligible. More importantly, when h is around $0.4\mu\text{m}$, I would expect the cells to actually touch the surface, so that the main origin of the torque would be the friction between the cell body and the gel: the cells would then reorient much like a rotating pen does when its tip is touching a piece of paper. Can the authors rule out this effect? Finally, I am not familiar with slender body theory: does it rely on far field hydrodynamics? If so, isn't this approximation violated when the cell is so close to the edge drop & to the gel surface? If the full hydrodynamic flow is accounted for, then it would be a good idea to indicate so for the uneducated reader (or referee).

Response:

In the wall-colliding case described by the reviewer, the wall is perpendicular to where the cells go, whereas in our case the main source of hydrodynamic interactions is on a surface parallel to the cells.

To rule out the possibility that the source or reorienting torque come from friction between cell body and the substrate, we may consider a cell swimming above an infinite surface. The cell body rotates CW because the flagellar bundle rotates CCW; the friction between the cell body and the substrate would then cause it to rotate CCW, but not CW as we observed. So in our case the reorienting torque must be acting on flagellar bundle and mediated through the fluid.

Regarding the slender body theory, yes our model takes into account full hydrodynamic interactions with the surface (except for lubrication when the helix is all but touching the surface). We have clarified this point in revised Methods section (lines 463-464).

6. Beyond these precise points, I found frustrating that the authors present a phenomenon, but do not discuss the conditions for its occurrence (i.e., the control parameters): would the presence of any circular boundary lead to this CW motion? Does it depend on the contact angle of the drop in which the bacteria live? What controls the widths of both layers? What happens precisely for the smooth swimmers case reported by the authors, which does not show such driven CW motion?

Response:

We apologize for the lack of discussions on these points in the previous manuscript.

(a). On the conditions of motile ring occurrence: In short, not all circular boundary would lead to the, and it does depend on the contact angle of the liquid drop (or more precisely, the width of wetting film at the edge of liquid drops). We have carefully monitored the timing of motile ring emergence in naturally developed colonies. We measured the collective speed of bacteria at colony edge over a time course of 7 hours starting from ~16 hours after colony inoculation, via long-term time lapse microscopy. At the early stage of colony development, cells adapt to the surface environment, extract water from the substrate, and become able to move on agar surface. At this stage the motion of cells displayed certain degree of ordering (as shown by the non-zero mean tangential speed of ~4 micron/s prior to Time=0 min in Fig. R4), presumably due to the CW bias of cell reorientation when colliding with the edge of colony (Fig. 3 in main text). This stage lasts for several hours until ~20 hr after colony inoculation, at which time a highly organized outer motile ring very rapidly emerges at the colony edge: The full development is completed within 10 min, as shown by the sharp transition of mean tangential speed at T=0 min in Fig. R4. The results are incorporated in lines 145-162 and Fig. 1F of the revised manuscript.

The result presented in Fig. R4 suggests that either the physiology of cells in the colony or the physicochemical conditions of the colony just reached an appropriate state for motile ring emergence immediately prior to the transition point at T= 0 min. As cells already started moving many hours before this transition point, the physiological state of cells must be similar before and after the transition; so it is more likely that the physicochemical conditions of the colony, such as water content or surface tension, are playing the key role. These factors are difficult to manipulate in naturally developed colonies because the process of water extraction and the molecular details of biosurfactant synthesis are unclear (see the references Hennes et al. 2017 and Zhang et al. 2010 listed below). Nonetheless, when we prepare artificial colonies (i.e. suspension drops), we notice that appropriate amount of surfactant (Tween 20; ~0.002% wt/wt) has to be supplemented to the medium. This fact suggests that surface tension of the colony has to be sufficiently low, such that a thin wetting film ~1 micron in thickness can form at the colony edge to support 2D motion of cells. So we believe surface tension is a main determinant for motile rings to emerge in naturally developed colonies. On the other hand, water content of the colony may be an important factor for the timing of onset of motile rings as well. We varied agar concentration and environmental humidity when growing *P. mirabilis* colonies and found that motile rings appeared earlier on agar plates with higher agar concentrations, and appeared earlier in less humid environment (but the humidity cannot be lower than 47-55%, otherwise the colony swarms quickly) (see Table R1). The environmental humidity may affect the water extraction process of the colony or the production of surfactant-like substances. However, the underlying mechanism is not clear to us and it is worth further study. This discussion is incorporated in lines 145-162 and Table S1 of the revised manuscript.

References:

1. Hennes, M., Tailleur, J., Charron, G., & Daerr, A. (2017). Active depinning of bacterial droplets: The collective surfing of *Bacillus subtilis*. *Proc. Natl. Acad. Sci. USA*. 114 (23): 5958-5963.
2. Zhang, R., Turner, L., & Berg, H. C. (2010). The upper surface of an *Escherichia coli* swarm is stationary. *Proc. Natl. Acad. Sci. USA*, 107, 288–290.

(b). On the widths of both motile layers: Please see Response to Point 4 (part (b)) above.

(c). On the behavior of smooth swimmers: We characterized the motion pattern of cells at the colony edge of *B. subtilis* smooth-swimming mutant (Fig. R5). Indeed, jammed clusters frequently formed at the edge (see Movie S5), presumably because these smooth-swimming cells were not able to switch flagellar rotation direction autonomously and tended to get stuck to each other during collisions. As a result of these jammed clusters, the orientation of cells at the colony edge remained disordered (Fig. R5, panel A), and unidirectional motion could not develop there, despite a weak CW bias in the average speed of cells (Fig. R5, panel B). This result is now presented in lines 178-182 and supplemented as Fig. S2 and Movie S5 in revised manuscript.

Fig. R4. Dynamics of motile ring emergence during colony growth. The mean tangential speed of bacteria (computed by optical flow analysis with phase contrast images) in the 10- μ m-wide outmost rim of colonies (i.e. the region of outer motile ring) is plotted against time. Time = 0 min is chosen at the onset of collective motion with high polar order and it corresponds to \sim 20 hr colony growth.

Fig. R5. Motion pattern of smooth swimming *B. subtilis* at colony edge. (A) Phase contrast image of the colony edge of smooth swimming *B. subtilis* mutant (DK2178). Scale bar, 20 μ m. (B) The mean tangential speed of smooth swimming *B. subtilis* cells (computed by optical flow analysis with phase contrast images) is plotted against the distance from colony edge. Positive value of speed indicates motion along CW direction, i.e. along +Y axis in the coordinate system specified in panel A, and X = 0 is set at the position of colony edge. Also see Movie S5.

A

The effect of LB agar concentration on the growth of *P. mirabilis* colonies with relative humidity 85%RH

	LB concentration (%)			
	0.6	1.0	1.5	2.0
Sessile colony with motile rings	Yes	Yes	Yes	Yes
Initiation of motile rings (hour)	19 ± 1	19 ± 1	17 ± 1	17.5 ± 0.5
Swarming	Yes	Yes	Yes	Yes
Initiation of swarming (hour)	> 24	> 24	> 24	> 24

B

The effect of relative humidity on the growth of *P. mirabilis* colonies with LB agar concentration 0.6%.

	Relative humidity (%RH)					
	97.0	85.0	60.4	55.0	47.4	42.0
Sessile colony with motile rings	Yes	Yes	Yes	Yes	n.a.	n.a.
Initiation of motile rings (hour)	19 ± 1	19 ± 1	15.5 ± 0.5	14.5 ± 0.5	n.a.	n.a.
Swarming	Yes	Yes	Yes	Yes	Yes	Yes
Initiation of swarming (hour)	> 24	> 24	> 24	> 24	4.5 ± 0.5	4.5 ± 0.5

Table R1. (A) The effect of LB agar concentration on the growth of *P. mirabilis* colonies cultured at 85% relative humidity. (B) The effect of relative humidity on the growth of *P. mirabilis* colonies cultured on 0.6% LB agar.

7. More minor points: I found the connection to active rods (lines 321-323) surprising since the dense lanes described there seem to be lacking here, and the order seems to have a completely different origin (alignment upon collision vs presence of an ordering boundary). I was also surprised to read about the lack of self-organization of swarming cells (lines 347-348) since I thought that the swarming of *B. subtilis* was actually explained precisely by a complex self-organization (see, e.g., Hamze et al, *microbiology* 157:2456-246 (2011)).

Response:

(a). The connection to modeling results of active rods (Ginelli et al., *Physical Review Letters* 104: 184502, 2010) was made in terms of the nematic order in cells' moving directions within the inner motile ring. The shear-induced depletion mechanism (see Response to Point 4, part (b)) can only explain the accumulation of cells near the outer motile ring, but it cannot explain how the nematic order in cells' moving directions arises. This question is nontrivial, because

advection due to fluid flows generated by flagellated bacteria at confinement boundary was shown to stabilize *unidirectional* collective motion (in which cells align their orientations nematically but move *unidirectionally*) (Lushi et al., PNAS 111: 9733-9738, 2014), and nematically ordered collective motion (in which cells move *bi-directionally* in parallel with each other as seen in our phenomenon) was not reported. We thereby make connection to Ginelli et al.'s modeling results in Discussion section, while pointing out the importance of hydrodynamic interaction in addition to nematic alignment between cells. This part of discussion has been revised to clarify these points.

(b). We thank the reviewer for pointing out the potentially misleading statement. By saying that “cells in bacterial swarms do not display any self-organization” we referred to the order of cells’ orientations and moving directions. The macroscopic population structure of swarming colonies as reported by Hamze et al. (Microbiology 157:2456-246, 2011) certainly represents a form of self-organization. Nonetheless, at mesoscales, the motion of cells in bacterial swarms is rather erratic and does not display any stable organization in cellular orientations or moving directions; one can only observe transient jets and vortices on the order of tens of microns lasting for a fraction of a second. For example, please see a movie of *E. coli* swarm edge in the following link:

http://www.rowland.harvard.edu/labs/bacteria/movies/showmovie.php?mov=An_E_coli_swarm;

and a movie of *B. subtilis* swarm here:

http://www.phy.cuhk.edu.hk/yiwu/Gallery/B.subtilis_swarm_edge.html. We have revised the sentence to be more clear and specific: “cells at the edge of bacterial swarms do not display stable ordering of orientations or moving directions despite the presence of transient jets and vortices lasting for a fraction of a second”.

Reviewers' comments:

Reviewer #1 (Remarks to the Author):

In the revised manuscript, the authors have improved the description of motile ring formation. They documented the environmental conditions (humidity, agar concentration), which are critical for motile ring formation and they specified the timing of its development after colony inoculation on the agar plate. In artificial droplets, they performed additional experiments showing that the speed of the outer ring displays a sharp transition with the bacterial density. They also additionally showed that the width of the inner motile ring scales linearly with the speed of the outer ring. Finally, they showed that large fluorescently labeled Dextran molecules (2000kDa) alike fluorescent microspheres are advected by the circular flow generated by swimming bacteria. And, they better described crack formation by measuring their mean radial extension and report the timing of their formation relative to the onset of outer ring formation. Although the authors report an original observation, the underlying physics is very confusing and the interpretations of the phenomenon are not supported by experimental data. These points are developed below. For these reasons, I do not recommend the article for publication in Nature Communications.

1. The authors claim that the motion of bacteria is nematic in the inner ring. However, Fig1C and 2B display a highly disordered velocity field, i.e. showing no preferential direction. In addition, the distribution reported in Fig1E only investigates the projection of the velocity parallel to the colony edge.

2. The authors use the term phase-separation and self-organization. However, it seems that the organization of the motile rings is essentially driven by the boundary. Accordingly, the linear relationship between the velocity of the outer ring and the width of the inner ring (Fig2E) simply reflects the fact that the width of the boundary layer scales with the speed of the boundary.

3. The authors mention shear-induced depletion to explain the organization of the inner ring. However, they never measured the radial profile of bacterial density to prove that low shear region are indeed depleted from bacteria. In addition, according to the authors, the height of the liquid film in which bacteria are confined varies by a factor 2 from the outer ring and the inner ring (Line 125 and Line 131), which complicates a bit the interpretations as this can generate density fluctuations independent of the shear profile.

4. The authors claim that motile rings provide the colony with long-range transport. However, nutrients are mainly depleted in the center of the colony during growth, while fresh nutrients are always available at the periphery of the colony. So, we can question the biological relevance of a directed transport along the boundary of the colony, which does not deliver nutrients at the center of the colony. Finally, the authors report in their response to my comments (response to point 5 at page7) that calcein, a small fluorescent probes (1kDa) is mostly diffusing and is poorly transported by advection through the flow generated by swimming bacteria.

5. In the manuscript the authors advance contradictory arguments. For instance they show Fig2F that the velocity of the outer ring strongly depends on bacterial density. However, they write in lines 154-158 that the sudden change in the velocity of the outer ring is due to change in the physio-chemical condition of the colony because the cell density is constant within this time interval.

Eventually, I still have minor concern regarding the notation. I advise to use V_{ring} when the velocity is measured by optical flow and V_{cell} , when velocity is extracted from single cell trajectories. Those notations should be consistent in the main text, the figures and the captions. And last, the size of the window used in optical flow is not mentioned in the methods nor in the supplementary materials nor in the text.

Reviewer #2 (Remarks to the Author):

The authors have thoroughly rewritten their paper and seriously considered all my comments. I think the manuscript has improved a lot and I recommend its publication in Nature Communication.

Response to the comments of Reviewer #1:

Although the authors report an original observation, the underlying physics is very confusing and the interpretations of the phenomenon are not supported by experimental data. These points are developed below.

Response: We thank the reviewer for taking the time and efforts to review our manuscript again. We apologize for any confusion and presentation of inadequate data. While the primary focus of our manuscript is to report and quantitatively characterize the novel phenomenon, we have clarified the potentially confusing parts of the manuscript in order to provide a more accurate picture of the underlying physics. Please see the specific points below.

1. The authors claim that the motion of bacteria is nematic in the inner ring. However, Fig1C and 2B display a highly disordered velocity field, i.e. showing no preferential direction. In addition, the distribution reported in Fig1E only investigates the projection of the velocity parallel to the colony edge.

Response:

As cells are moving bi-directionally in the inner motile ring, their average velocities are rather small and fluctuate in space. For this reason, the collective velocity fields (in which every velocity vector represents the average motion of many cells) in Fig. 1C and Fig. 2B appear disordered, and therefore the collective velocity field is not an appropriate proxy to infer nematic order of cellular motion. Instead, the inset figures in Fig. 1E and Fig. 2D in main text presenting the velocity direction distribution of individual bacteria in the inner motile ring were meant to indicate the nematic order: The two inset figures show that the moving directions are preferentially parallel to the boundary bi-directionally.

To make this point more clear, here we present the same single-cell velocity data in Fig. 1E and Fig. 2D in an alternative way, using the angular probability distribution of velocity direction and the directional dependence of average speed (*i.e.* velocity magnitude) (Fig. R1). As shown in Fig. R1-A (for inner motile ring in a naturally grown *P. mirabilis* colony) and Fig. R1-B (for inner motile ring in a *P. mirabilis* suspension drop), the probability distribution of cells' moving direction in the inner motile ring is centered around 90 degrees and 270 degree, *i.e.* preferentially parallel to the boundary bi-directionally. In addition, the directional dependence of average speed is also anisotropic as shown in Fig. R1-C and Fig. R1-D, which complement the probability distributions of cell speed presented in Fig. 1E and Fig. 2D in main text. Please note that the directional dependence of average speed is more isotropic in suspension drops (Fig. R1-C) than that in natural colonies (Fig. R1-D); this is because there is no sessile part in suspension drops and thus cells are able to move between the inner motile ring and the dilute phase in all directions without much obstruction (or reduction of speed). Fig. R1 and the above discussions are supplemented in Fig. S1 in our revised manuscript.

Fig. R1. (A,B) Angular probability distribution of single-cell velocity direction in the inner motile ring of a naturally grown *P. mirabilis* colony (panel A) and of a *P. mirabilis* suspension drop (panel B), respectively. The velocity direction is represented by angles ranging from 0° to 360° . 90° and 270° correspond to +Y (clockwise along the edge) and -Y (counterclockwise along the edge) directions in the coordinate system specified in main text Fig. 1B and Fig. 2A, respectively. The radii of colored circular sectors represent probability density. (C,D) Directional dependence of average speed in the inner motile ring of a naturally grown *P. mirabilis* colony (panel C) and of a *P. mirabilis* suspension drop (panel D), respectively. The radii of colored circular sectors represent the magnitude of speed, with the scale indicated in the plots.

2. *The authors use the term phase-separation and self-organization. However, it seems that the organization of the motile rings is essentially driven by the boundary. Accordingly, the linear relationship between the velocity of the outer ring and the width of the inner ring (Fig2E) simply reflects the fact that the width of the boundary layer scales with the speed of the boundary.*

Response:

To clarify the self-organization nature of the CW outer motile ring, we had tuned the cell density in suspension drops of *P. mirabilis* and found that the order of cellular motion in the 10- μm -wide outmost rim of the drop depends on cell density (Fig. 2F of main text). At low densities, cellular motion in this rim does not display polar order, despite a weak CW bias (mean tangential speed $\sim 2 \mu\text{m/s}$) due to the CW bias of cell reorientation upon contact with drop edge; beyond a critical cell density (cell occupation ratio ~ 0.4), cellular motion experiences a sharp transition to maximal polar order (mean tangential speed $\sim 12 \mu\text{m/s}$), and a highly ordered CW motile ring emerges. This result demonstrates that the emergence of a highly ordered outer motile ring is a self-organized collective effect driven by cell-cell interactions, but not by cell-boundary interactions. On the other hand,

the CW chirality of outer motile rings presumably results from the CW bias of single-cell reorientation upon contact with drop edge; therefore, only the CW chirality of outer motile ring is driven by cell-boundary interactions.

We agree with the reviewer that the linear relationship in Fig. 2E simply reflects the fact that the width of the boundary layer scales with the speed of the boundary. However, this linear scaling of the width of inner motile ring is non-trivial. It is not clear to us why that linear scaling arises and why this inner penetration length should scale linearly with the applied stress due to the outer motile ring. This dependence indicates a nonlinear problem and many factors are probably important: steric, hydrodynamic, and also the fact that the density of cells might change as one moves away from the boundary.

Indeed, if we make analogy between the inner motile ring and boundary layer (BL) seen in passive systems, the scaling of BL is not linear in a variety of well-known examples: It is not linear in classical high-Re flows [typically in that case the BL size goes like the inverse square root of U]; in Stokes first problem where a moving wall (at speed U) entrains a viscous fluid on top of it, the width of the entrained fluid is not proportional to U and is in fact independent of U (the size of the BL grows in time diffusively where the kinematic viscosity is the diffusion constant; see Leal's textbook for example); for a heterogeneous fluid in porous medium, the penetration length also does not scale like U but is set by the pores/microstructure of the fluid [see Brinkman equations for example]. A better understanding of the width of both motile rings as well as the stability of inner motile ring may require modeling of dense suspensions of swimmer that takes into account both hydrodynamic and steric interactions, which would be out of the scope of the current paper.

3. The authors mention shear-induced depletion to explain the organization of the inner ring. However, they never measured the radial profile of bacterial density to prove that low shear region are indeed depleted from bacteria. In addition, according to the authors, the height of the liquid film in which bacteria are confined varies by a factor 2 from the outer ring and the inner ring (Line 125 and Line 131), which complicates a bit the interpretations as this can generate density fluctuations independent of the shear profile.

Response:

We have measured the radial profile of bacterial density in suspension drops and the result is shown in Fig. R2-A below. Near the transition region between the dilute phase and the inner motile ring (at ~40 micron from the edge), the surface cell density experienced a sharp increase when moving towards the edge. This increase of surface cell density cannot be attributed to the variation of the height of liquid film, since the fluid height in the dilute phase is comparable or greater than that in the inner motile ring; indeed, the increase of surface cell density reflects the increase of volume cell density and proves that cells accumulate towards the edge from the dilute phase. (Please note that the difference in fluid height or cell density between the outer ring and the inner ring is irrelevant for the interpretation of shear-induced depletion mechanism; we only need to consider the difference between the inner ring and dilute phase). The region with cell accumulation (~10-40 micron from the edge, excluding the outer motile ring) corresponds to the region of inner motile ring (Fig. 2A in main text), and it coincides with

the region with horizontal shear shown in Fig. R2-B (*i.e.* Fig. S7 of the manuscript). Therefore the result presented in Fig. R2-A provides another piece of evidence supporting that fluid shear generated by cells in the outer motile ring helps attract cells from the dilute phase towards the colony edge and contributes to the formation of inner motile ring. We have included Fig. R2-A in the revised manuscript as Figure S9 and incorporated the above discussions in lines 352-362 of main text.

Fig. R2. (A) Surface cell density plotted against the distance from the edge of a *P. mirabilis* suspension drop. The surface cell density (number of cells per unit area of substrate surface) is represented by fluorescence intensity, as all cells in the suspension drop were fluorescently labeled (expressing GFP). Fluorescence intensity of cells was measured when the CW outer motile ring and CCW inner motile ring both appeared and had stabilized at the edge. (B) Mean tangential flow speed plotted against the distance from the edge of a *P. mirabilis* suspension drop. Positive value of speed indicates motion along CCW direction (see the caption of Fig. S7 in manuscript for more information). In both panels $X = 0$ is set at the position of the edge of the suspension drop.

4. The authors claim that motile rings provide the colony with long-range transport. However, nutrients are mainly depleted in the center of the colony during growth, while fresh nutrients are always available at the periphery of the colony. So, we can question the biological relevance of a directed transport along the boundary of the colony, which does not deliver nutrients at the center of the colony. Finally, the authors report in their response to my comments (response to point 5 at page7) that calcein, a small fluorescent probes (1kDa) is mostly diffusing and is poorly transported by advection through the flow generated by swimming bacteria.

Response:

For bacterial colonies grown in nutrient environments with rotational symmetry, we agree with the reviewer that it is more useful to transport nutrients into the center than to transport nutrients around the colony periphery. However, bacteria communities in natural and clinical settings normally grow in anisotropic, structured environments with heterogeneous nutrient or chemical distribution. In such environments, bacterial communities do not have rotational symmetry and the self-organization of motile subpopulation may occur in various locations not limited to the colony edge; thus the long-range directed flows enabled by motile-cell self-organization may efficiently redistribute nutrients and signaling molecules across different regions of heterogeneous bacterial communities. This directed transport in heterogeneous environments is most effective for slowly-diffusing substances, such as high-molecular-weight polymeric metabolites and membrane vesicles of size ~ 0.1 micron that encapsulate nutrients or signaling molecules. These points of biological relevance were clarified in lines 436-450 in main text.

5. In the manuscript the authors advance contradictory arguments. For instance they show Fig2F that the velocity of the outer ring strongly depends on bacterial density. However, they write in lines 154-158 that the sudden change in the velocity of the outer ring is due to change in the physio-chemical condition of the colony because the cell density is constant within this time interval.

Response:

We apologize for not explaining the difference between Fig. 1F and Fig. 2F of main text clearly. As the reviewer pointed out and as shown in Fig. 1F, at ~ 20 hr after colony inoculation (i.e. $T=0$ min in Fig. 1F), a highly ordered outer motile ring emerges rapidly at the colony edge. On the other hand, Fig. 2F reveals that the emergence of the highly ordered outer motile ring is a self-organized collective effect arising from cell-cell interactions (also see response to Point 2 above). Fig. 1F and Fig. 2F reveals the two essential requirements of forming a *highly ordered* outer motile ring: (1) Appropriate physicochemical conditions of the colony, presumably water content and surface tension, that permit the formation of a thin wetting film $\sim 1-2$ μm in thickness at the colony edge in order to support 2D motion of cells; (2) Sufficiently high cell density that allows for intensive cell-cell interactions. As we suggested in main text, this first requirement was not satisfied in naturally developed *P. mirabilis* colonies prior to $T=0$ min in Fig. 1F, although the second requirement had already been met. By contrast, the first requirement was always satisfied in suspension drops of *P. mirabilis*, because we supplied exogenous surfactant (Tween 20) at appropriate concentrations to ensure that a thin wetting film $\sim 1-2$ micron in thickness could form at the edge of the liquid drop and could support 2D motion of cells. We have revised the main text (Lines 227-236) and clarified these points to avoid confusion.

6. Eventually, I still have minor concern regarding the notation. I advise to use V_{ring} when the velocity is measured by optical flow and V_{cell} , when velocity is extracted from single cell trajectories. Those notations should be consistent in the main text, the figures and

the captions. And last, the size of the window used in optical flow is not mentioned in the methods nor in the supplementary materials nor in the text.

Response:

We thank the reviewer for making this suggestion. We have changed to use V_{ring} (when referring to collective velocity measured by optical flow) and V_{cell} (when referring to velocities obtained by single cell tracking) throughout the text and figures. We also revised the supplementary methods (lines 190-201) to include a more detailed description of optical analysis as follows:

“The velocity field of cells’ collective motion was obtained by performing optical flow analysis based on microscopy Movies using the built-in functions of MATLAB. Prior to computing the optical flow fields, the images were first smoothed to reduce noise by convolution with a Gaussian kernel of standard deviation 1. The optical flow field for any two consecutive video frames was computed using the Horn-Schunck algorithm (maximum iteration number, 128; smoothness parameter, 1) and then smoothed by local averaging. The grid size of the optical flow field was 1 pixel x 1 pixel and the initial value of optical flow vectors was set to zero. The obtained optical flow fields were used to compute the collective speed profile in Fig. 1D and Fig. 2C. To visualize the collective velocity field (Fig. 1C and Fig. 2B), the optical flow fields were coarsened to a grid size of $5.2 \mu\text{m} \times 5.2 \mu\text{m}$. The results were insensitive to different parameters of smoothing. ”

REVIEWERS' COMMENTS:

Reviewer #1 (Remarks to the Author):

The authors have thoroughly revised their paper and addressed my comments. I recommend its publication in Nature Communications.